# Task Arithmetic in the Tangent Space:
# Improved Editing of Pre-Trained Models

**Guillermo Ortiz-Jimenez**[*]
EPFL, Lausanne, Switzerland
guillermo.ortizjimenez@epfl.ch

**Alessandro Favero**[*]
EPFL, Lausanne, Switzerland
alessandro.favero@epfl.ch

**Pascal Frossard**
EPFL, Lausanne, Switzerland
pascal.frossard@epfl.ch

## Abstract

Task arithmetic has recently emerged as a cost-effective and scalable approach to edit pre-trained models directly in weight space: By adding the fine-tuned weights of different tasks, the model's performance can be improved on these tasks, while negating them leads to task forgetting. Yet, our understanding of the effectiveness of task arithmetic and its underlying principles remains limited. We present a comprehensive study of task arithmetic in vision-language models and show that *weight disentanglement* is the crucial factor that makes it effective. This property arises during pre-training and manifests when distinct directions in weight space govern separate, localized regions in function space associated with the tasks. Notably, we show that fine-tuning models in their tangent space by linearizing them amplifies weight disentanglement. This leads to substantial performance improvements across multiple task arithmetic benchmarks and diverse models. Building on these findings, we provide theoretical and empirical analyses of the neural tangent kernel (NTK) of these models and establish a compelling link between task arithmetic and the spatial localization of the NTK eigenfunctions. Overall, our work uncovers novel insights into the fundamental mechanisms of task arithmetic and offers a more reliable and effective approach to edit pre-trained models through the NTK linearization.

## 1 Introduction

Pre-trained models play a pivotal role in contemporary machine learning systems. However, to enhance performance on downstream tasks [38, 39, 92], align them with human preferences [31, 50, 64, 74], and increase robustness [63, 76, 87], they often necessitate further editing. Traditional model editing methods rely on costly joint fine-tuning across multiple tasks [92] and human-feedback [64], which constrain scalability and democratization. Furthermore, enhancing downstream task performance typically degrades the model's pre-training performance or *zero-shot* accuracy [28, 58, 87].

Recent research has introduced cost-effective and scalable model editing techniques that try to preserve the pre-trained model behavior by acting on the model weights through *task arithmetic* or weight interpolation techniques [3, 23, 27, 38–40, 46, 57, 71, 72, 79, 86, 87, 89], thus circumventing expensive joint fine-tuning over multiple tasks. These methods hinge on the observation that arithmetic operations between fine-tuned weights often produce analogous functional behaviors [39]. For example, summing the relative weight components of a model between pre-training and fine-tuning

---

[*]Equal contribution.

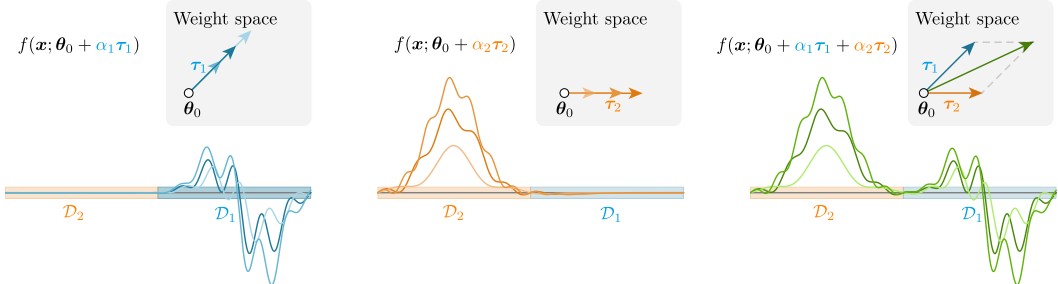

Figure 1: **Illustration of weight disentanglement**, where distinct directions in the weight space, $\boldsymbol{\tau}_t$, are associated with localized areas of the input space, $\mathcal{D}_t$. This allows a model, $f$, to manipulate these areas independently by adding linear combinations of $\boldsymbol{\tau}_t$'s to a pre-trained checkpoint $\boldsymbol{\theta}_0$.

on two separate tasks results in a new multi-task model with improved performance on both tasks. Similarly, subtracting a task's relative component can lead to the model forgetting that task.

Despite these advancements, the understanding of task arithmetic's underlying principles and its general effectiveness remains limited. Specifically, a comprehensive understanding of how these techniques affect a model's internal representations and the necessary conditions to make it reliable is lacking. This knowledge gap can undermine the adoption of these techniques, as it erodes their trustworthiness in real-world applications. In addition, reducing this gap could help us improve them even further.

To address these challenges, we present a systematic study of task arithmetic in contrastively pre-trained vision-language models (*i.e.*, CLIP [69]), offering novel insights into its underlying mechanisms and introducing new approaches which enhance the performance of pre-trained models edited through task arithmetic. Specifically, we probe the hypothesis presented in Wortsman et al. [87] that task arithmetic is possible thanks to the fact that models inherently operate in a linear regime, where their behavior is dictated by the finite-width neural tangent kernel (NTK) [16, 41].

Our study reveals that linearized CLIP models exhibit significantly improved task arithmetic performance with respect to their nonlinear counterparts (see Tables 1 and 2), but also that the NTK cannot fully account for the task arithmetic abilities of pre-trained non-linear models. Indeed, we show that the sole requirement for task arithmetic is actually *weight disentanglement*, where distinct directions in weight space correspond to changes of the network in disjoint regions of the input space (see Figure 1). This allows a model to perform task arithmetic by independently manipulating these weight directions.

Notably, we show that fine-tuning models in their tangent space by linearizing them amplifies weight disentanglement, leading to substantial performance improvements across multiple task arithmetic benchmarks and models. However, although weight disentanglement is stronger in the tangent space, it is also present in non-linear models. We demonstrate that weight disentanglement of semantically meaningful tasks is an emergent property of pre-training, as it is absent at random initialization.

In particular, our main contributions are as follows:

- We formalize the notion of task arithmetic introduced in Ilharco et al. [39] as Property 1, allowing us to reason quantitatively about it.
- We show that task arithmetic in non-linear models cannot be explained by their NTK, and introduce the concept of weight disentanglement as the necessary condition to enable it.
- We propose to linearize models as a way to enhance weight disentanglement and improve task arithmetic. Doing so, we achieve up to $5.8$ points more and $13.1$ points less in accuracy on task addition and task negation, respectively, on several vision-language benchmarks.
- We link weight disentanglement in linearized models to spatial localization of the kernel eigenfunctions and validate this prediction numerically in pre-trained transformer models.
- Finally, we show that weight disentanglement is an emergent property of pre-training.

Overall, our work delivers new insights into the fundamental mechanisms of task arithmetic, facilitating more reliable and scalable model editing. Our findings suggest that linearized fine-tuning of pre-trained models warrants further investigation, with the potential for substantial impact on effective model editing. These insights can foster the development of more efficient and precise model editing techniques, empowering practitioners to adapt pre-trained models to a broader range of tasks.

## 2 Notation and problem statement

Let $f : \mathcal{X} \times \Theta \to \mathcal{Y}$ be a neural network taking inputs $\boldsymbol{x} \in \mathcal{X}$ and parameterized by a set of weights $\boldsymbol{\theta} \in \Theta$. We will assume $\mathcal{X} \subseteq \mathbb{R}^d$, $\Theta \subseteq \mathbb{R}^m$ and $\mathcal{Y} \subseteq \mathbb{R}^c$. Consider $T$ tasks, with every task $t$ consisting of a triplet $(\mathcal{D}_t, \mu_t, f_t^\star)$ where $\mathcal{D}_t \subseteq \mathcal{X}$ is a data support (*e.g.*, ImageNet [21] images), $\mu_t$ an input distribution such that $\operatorname{supp}(\mu_t) = \mathcal{D}_t$, and $f_t^\star : \mathcal{D}_t \to \mathcal{Y}$ a target function (*e.g.*, labels). In practice, each task is identified with a training set $\{(\boldsymbol{x}_\nu, f_t^\star(\boldsymbol{x}_\nu))\}_{\nu \in [n_t]}$ with $\boldsymbol{x} \sim \mu_t$, that is used to fine-tune the models starting from the pre-trained weights $\boldsymbol{\theta}_0$ and obtain the fine-tuned weights $\boldsymbol{\theta}_t^\star$.

**Task arithmetic.**    Let the *task vector* of task $t$ be the difference between the fine-tuned and the pre-trained weights, *i.e.*, $\boldsymbol{\tau}_t = \boldsymbol{\theta}_t^\star - \boldsymbol{\theta}_0$. The following property formalizes the notion of task arithmetic introduced in Ilharco et al. [39], where the authors observed that the accuracies of pre-trained models on different datasets can be modified independently through the addition or removal of task vectors.

**Property 1** (Task arithmetic). *Consider a set of task vectors $\mathcal{T} = \{\boldsymbol{\tau}_t\}_{t \in [T]}$ with associated non-intersecting task supports $\mathcal{D} = \{\mathcal{D}_t \subset \mathcal{X}\}_{t \in [T]}$, i.e., $\forall t, t'$, if $t \neq t'$ then $\mathcal{D}_t \cap \mathcal{D}_{t'} = \varnothing$. We say a network $f$ satisfies the task arithmetic property around $\boldsymbol{\theta}_0$ with respect to $\mathcal{T}$ and $\mathcal{D}$ if*

$$f\left(\boldsymbol{x}; \boldsymbol{\theta}_0 + \sum_{t=1}^T \alpha_t \, \boldsymbol{\tau}_t\right) = \begin{cases} f(\boldsymbol{x}; \boldsymbol{\theta}_0 + \alpha_t \, \boldsymbol{\tau}_t) & \boldsymbol{x} \in \mathcal{D}_t \\ f(\boldsymbol{x}; \boldsymbol{\theta}_0) & \boldsymbol{x} \notin \bigcup_{t=1}^T \mathcal{D}_t \end{cases} \tag{1}$$

*with $(\alpha_1, \ldots, \alpha_T) \in \mathcal{A} \subseteq \mathbb{R}^T$.*

In short, a model satisfies Property 1 if adding $\boldsymbol{\tau}_t$ does not modify the output of the model outside $\mathcal{D}_t$.

**Neural tangent kernel.**    Around the initialization weights $\boldsymbol{\theta}_0$, a neural network can be approximated with a first-order Taylor expansion:

$$f(\boldsymbol{x}; \boldsymbol{\theta}) \approx f(\boldsymbol{x}; \boldsymbol{\theta}_0) + (\boldsymbol{\theta} - \boldsymbol{\theta}_0)^\top \nabla_{\boldsymbol{\theta}} f(\boldsymbol{x}; \boldsymbol{\theta}_0). \tag{2}$$

This approximation is equivalent to a kernel predictor with a kernel known as the *neural tangent kernel* (NTK) [41], $k_{\mathrm{NTK}}(\boldsymbol{x}, \boldsymbol{x}') = \nabla_{\boldsymbol{\theta}} f(\boldsymbol{x}; \boldsymbol{\theta}_0)^\top \nabla_{\boldsymbol{\theta}} f(\boldsymbol{x}'; \boldsymbol{\theta}_0)$, and defines a neural tangent space in which the relationship between weights and functions is linear. Remarkably, as the network width approaches infinity, Eq. (2) becomes exact and remains valid throughout training [4, 41, 45].

However, this linear approximation is often invalid at finite widths, as the evolution of parameters during training is inadequately captured by Eq. (2). In such cases, training occurs in a *non-linear regime*. Conversely, often during fine-tuning, parameter evolution in many pre-trained models is frequently minimal, meaning that training does not exit the tangent space and Eq. (2) closely approximates the network behavior [22, 53, 62, 90, 91]. In such cases, training occurs in a *linear regime*.

## 3 Task arithmetic is not a consequence of linear fine-tuning

The objective of this work is to understand the conditions that enable task arithmetic in deep neural networks. Previous studies hypothesized that task arithmetic results from fine-tuning in the linear regime [39, 86, 87], as linear weight combinations correspond to similar output function combinations. However, we will now demonstrate that CLIP models do not fine-tune in the linear regime and we therefore need other ways to explain task arithmetic.

In general, if a pre-trained network $f(\,\cdot\,; \boldsymbol{\theta}_0)$ demonstrates *kernel behavior* during fine-tuning – *i.e.*, fine-tuning occurs in the linear regime – the following property must be satisfied [53]:

**Property 2** (Post-hoc linearization). *The change in the network output after training can be approximated by its first-order Taylor expansion, i.e., $f(\boldsymbol{x}; \boldsymbol{\theta}^\star) - f(\boldsymbol{x}; \boldsymbol{\theta}_0) \approx (\boldsymbol{\theta}^\star - \boldsymbol{\theta}_0)^\top \nabla_{\boldsymbol{\theta}} f(\boldsymbol{x}; \boldsymbol{\theta}_0)$.*

In simple terms, the approximation of the network in the tangent space around initialization must hold after fine-tuning. To test this, we evaluate the performance of the *post-hoc* linearized version of $f$, $f_{\mathrm{lin}}$. That is, we apply the fine-tuned task vectors $\boldsymbol{\tau} = \boldsymbol{\theta}^\star - \boldsymbol{\theta}_0$ to the linear approximation of $f$ at $\boldsymbol{\theta}_0$, *i.e.*,

$$f_{\mathrm{lin}}(\boldsymbol{x}; \boldsymbol{\theta}_0 + \boldsymbol{\tau}) = f(\boldsymbol{x}; \boldsymbol{\theta}_0) + \boldsymbol{\tau}^\top \nabla_{\boldsymbol{\theta}} f(\boldsymbol{x}; \boldsymbol{\theta}_0), \tag{3}$$

and we check whether $f_{\mathrm{lin}}(\,\cdot\,; \boldsymbol{\theta}^\star)$ performs similarly to $f(\,\cdot\,; \boldsymbol{\theta}^\star)$[2].

---

[2]The code to reproduce our experiments can be found at `https://github.com/gortizji/tangent_task_arithmetic`.

Table 1: **Task addition.** Average absolute (%) and normalized accuracies (%) of different CLIP ViTs edited by adding the sum of the task vectors of 8 tasks. We report results for the non-linear and linearized models of Sections 3 and 5 normalizing performance by their single-task accuracies.

| Method | | ViT-B/32 | | ViT-B/16 | | ViT-L/14 | |
|---|---|---|---|---|---|---|---|
| | | Abs. ($\uparrow$) | Norm. ($\uparrow$) | Abs. ($\uparrow$) | Norm. ($\uparrow$) | Abs. ($\uparrow$) | Norm. ($\uparrow$) |
| Pre-trained | $f(\cdot\,;\boldsymbol{\theta}_0)$ | 48.4 | – | 55.2 | – | 64.4 | – |
| Non-lin. FT | $f(\cdot\,;\boldsymbol{\theta}_0+\boldsymbol{\tau})$ | 71.4 | 76.5 | 75.5 | 80.0 | 85.1 | 88.8 |
| Post-hoc lin. | $f_{\text{lin}}(\cdot\,;\boldsymbol{\theta}_0+\boldsymbol{\tau})$ | 57.1 | 81.9 | 65.0 | 85.2 | 75.2 | 90.0 |
| Linear. FT | $f_{\text{lin}}(\cdot\,;\boldsymbol{\theta}_0+\boldsymbol{\tau}_{\text{lin}})$ | **76.5** | **85.4** | **81.3** | **86.0** | **88.5** | **93.5** |

Table 2: **Task negation.** Minimum accuracy (%) of different CLIP ViTs edited by negating a task vector from a target task while retaining 95% of their performance on the control task. We report average performances over eight tasks on non-linear and linearized models as introduced in Sections 3 and 5.

| Method | | ViT-B/32 | | ViT-B/16 | | ViT-L/14 | |
|---|---|---|---|---|---|---|---|
| | | Targ. ($\downarrow$) | Cont. ($\uparrow$) | Targ. ($\downarrow$) | Cont. ($\uparrow$) | Targ. ($\downarrow$) | Cont. ($\uparrow$) |
| Pre-trained | $f(\cdot\,;\boldsymbol{\theta}_0)$ | 48.4 | 63.4 | 55.2 | 68.3 | 64.4 | 75.5 |
| Non-lin. FT | $f(\cdot\,;\boldsymbol{\theta}_0-\boldsymbol{\tau})$ | 24.0 | 60.7 | 19.2 | 64.6 | 18.0 | **72.5** |
| Post-hoc lin. | $f_{\text{lin}}(\cdot\,;\boldsymbol{\theta}_0-\boldsymbol{\tau})$ | 14.8 | 60.3 | **10.8** | **64.8** | 12.1 | 71.8 |
| Linear. FT | $f_{\text{lin}}(\cdot\,;\boldsymbol{\theta}_0-\boldsymbol{\tau}_{\text{lin}})$ | **10.9** | **60.8** | 11.3 | **64.8** | **7.9** | **72.5** |

The results in Figure 2 indicate that CLIP models do not exhibit a kernel behavior. Specifically, we fine-tune (FT) several CLIP pre-trained Vision Transformers (ViTs) [24] of different sizes following the same setup as Ilharco et al. [39] on 8 tasks: Cars [43], DTD [20], SUN397 [88], EuroSAT [33], GTSRB [80], MNIST [44], SVHN [60] and RESISC45 [15]. We observe that the single-task performance of $f_{\text{lin}}(\cdot\,;\boldsymbol{\theta}^\star)$ is significantly lower than that of $f(\cdot\,;\boldsymbol{\theta}^\star)$ for ViTs of all sizes. This *non-linear advantage* [26] is a clear sign that fine-tuning has not happened in a linear regime as expected by Wortsman et al. [87].

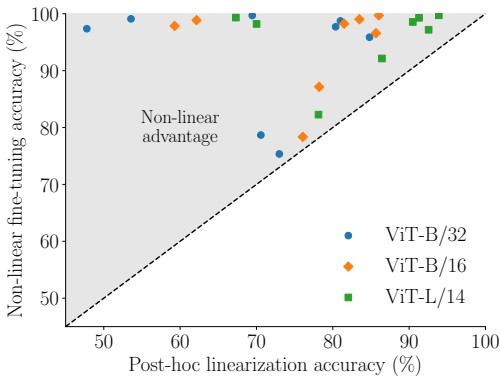

Figure 2: **Non-linear advantage.** Single-task accuracies of non-linearly fine-tuned models $f(\cdot\,;\boldsymbol{\theta}^\star)$ and their *post-hoc* linearization $f_{\text{lin}}(\cdot\,;\boldsymbol{\theta}^\star)$. Markers represent different ViTs.

Yet, this observation is not enough to rule out that task arithmetic can be explained by linearizing the network function. Indeed, even if the non-linear components are important for single-task performance, they might not be used during task arithmetic, which is the objective of this study. That is, the projection of $f$ onto the tangent space could be the only useful component.

We now show this is also not the case, as doing task arithmetic with the non-linearly fine-tuned task vectors over $f_{\text{lin}}$ significantly decreases performance. To show this, we employ the benchmark proposed in Ilharco et al. [39] to evaluate the task arithmetic ability of a pre-trained model, which consists of the 8 tasks described before and two sub-benchmarks:

1. **Task addition**: The sum of the task vectors $\boldsymbol{\tau} = \sum_t \boldsymbol{\tau}_t$ is added to a pre-trained checkpoint to produce a multi-task model. The success of this benchmark is measured in terms of the maximum average accuracy over the different tasks. Results are shown in Table 1.

2. **Task negation**: A task vector is subtracted from the pre-trained checkpoint to forget a task while retaining performance on a control task (ImageNet). The success of this benchmark is measured in terms of the maximum drop in accuracy on the forgetting task that retains the performance on the control task. Results are averaged over tasks and shown in Table 2.

To obtain the task vectors, we use the fine-tuned weights of the different ViTs from before, and use a single mixing coefficient $\alpha = \alpha_1 = \cdots = \alpha_T$ optimized separately for the non-linear and post-hoc linearized models to ensure a fair comparison. We provide all details of this experiment in Appendix A.

The results in Table 1 confirm that task arithmetic in CLIP models does not stem from the combination of their linear components only. Specifically, we observe a significant drop in absolute task addition accuracy in the *post-hoc* linearized models compared to the non-linear ones. This decrease in performance is consistent across tasks (see Appendix D.2) and highlights that task arithmetic in non-linear models leverages the non-linear components of $f$, as well.

Although these results reject the linear hypothesis, it is still remarkable that the post-hoc linearized models do better at task negation than the non-linear ones (see Table 2). Furthermore, even in task addition (see Table 1) they achieve higher normalized accuracies (see definition in Appendix A). Indeed, as we formalize in Section 4, this observation suggests that linearized models are more consistent with Property 1. In Section 5, we will use this fact to devise a new way to enhance task arithmetic.

## 4   Weight disentanglement

If the linear regime is not necessary to explain task arithmetic, what are the necessary conditions that allow it? In this section, we argue that the only necessary condition to perform task arithmetic with a model $f$ is that the model is *weight disentangled* with respect to the set of fine-tuning tasks.

**Property 3** (Weight disentanglement). *A parametric function $f : \mathcal{X} \times \Theta \to \mathcal{Y}$ is weight disentangled with respect to a set of task vectors $\mathcal{T} = \{\boldsymbol{\tau}_t\}_{t \in [T]}$ and the corresponding supports $\mathcal{D} = \{\mathcal{D}_t\}_{t \in [T]}$ if*

$$f\left(\boldsymbol{x}; \boldsymbol{\theta}_0 + \sum_{t=1}^{T} \alpha_t \boldsymbol{\tau}_t\right) = \sum_{t=1}^{T} g_t(\boldsymbol{x}; \alpha_t \boldsymbol{\tau}_t) + g_0(\boldsymbol{x}), \tag{4}$$

*where $g_t(\boldsymbol{x}; \alpha_t \boldsymbol{\tau}_t) = \mathbf{0}$ for $\boldsymbol{x} \notin \mathcal{D}_t$ and $t = 1, \ldots, T$, and $g_0(\boldsymbol{x}) = 0$ for $\boldsymbol{x} \in \bigcup_{t \in [T]} \mathcal{D}_t$.*

In essence, this definition captures the idea that the function $f$ can be decomposed as a sum of spatially-localized components, *i.e.*, vanishing outside a spatial region, whose functional variation is entirely captured by each $\boldsymbol{\tau}_t$ (see Figure 1). Moreover, it is trivial to see that satisfying weight disentanglement is equivalent to satisfying Property 1 on task arithmetic as one can always write Eq. (1) as

$$f\left(\boldsymbol{x}; \boldsymbol{\theta}_0 + \sum_{t=1}^{T} \alpha_t \boldsymbol{\tau}_t\right) = \sum_{t=1}^{T} f(\boldsymbol{x}; \boldsymbol{\theta}_0 + \alpha_t \boldsymbol{\tau}_t) \mathbb{1}(\boldsymbol{x} \in \mathcal{D}_t) + f(\boldsymbol{x}; \boldsymbol{\theta}_0) \mathbb{1}\left(\boldsymbol{x} \notin \bigcup_{t \in [T]} \mathcal{D}_t\right), \tag{5}$$

and identify $g_t(\boldsymbol{x}; \alpha_t \boldsymbol{\tau}_t) = f(\boldsymbol{x}; \boldsymbol{\theta}_0 + \alpha_t \boldsymbol{\tau}_t) \mathbb{1}(\boldsymbol{x} \in \mathcal{D}_t)$ and $g_0(\boldsymbol{x}) = f(\boldsymbol{x}; \boldsymbol{\theta}_0) \mathbb{1}(\boldsymbol{x} \notin \mathcal{D}_t)$. It is important to highlight, however, that this additive decomposition does not imply linearity, as the local functions $\{g_t\}_{t \in [T]}$ are not required to be linear with respect to the parameters.

Furthermore, note that weight disentanglement is a property of the predictors and not related to the performance on different tasks. That is, a model could be weight disentangled with respect to a set of task vectors and still perform poorly on a task, *e.g.*, if $f(\cdot\,; \boldsymbol{\theta}_0 + \alpha \boldsymbol{\tau})$ does not generalize for some $\alpha$. More generally, we can visualize the level of weight disentanglement of a model by measuring its discrepancy with Eq. (4). To do so, given two tasks, one can check the *disentanglement error* of a model,

$$\xi(\alpha_1, \alpha_2) = \sum_{t=1}^{2} \mathbb{E}_{\boldsymbol{x} \sim \mu_t} \left[\text{dist}\left(f(\boldsymbol{x}; \boldsymbol{\theta}_0 + \alpha_t \boldsymbol{\tau}_t), f(\boldsymbol{x}; \boldsymbol{\theta}_0 + \alpha_1 \boldsymbol{\tau}_1 + \alpha_2 \boldsymbol{\tau}_2)\right)\right], \tag{6}$$

where $\text{dist}$ denotes any distance metric between output vectors. As we are dealing with classification tasks, in what follows we use the prediction error $\text{dist}(y_1, y_2) = \mathbb{1}(y_1 \neq y_2)$ as the distance metric. In general, the smaller the value of $\xi(\alpha_1, \alpha_2)$ the more weight disentangled a model is at $(\alpha_1, \alpha_2)$.

Figure 3 displays the disentanglement error of a CLIP ViT-B/32 model concerning several task vector pairs. We observe that the CLIP model exhibits a minimal disentanglement error within a small region surrounding $\boldsymbol{\theta}_0$, which enables task arithmetic. However, for $\alpha_1, \alpha_2 > 1$, the error increases, indicating a high degree of interaction between tasks. This explains why task arithmetic performs better in a small neighborhood of $\boldsymbol{\theta}_0$ – task arithmetic is more effective when fine-tuning with small learning rates and few training steps [39] – with the optimal value of $\alpha$ typically being less than 1.

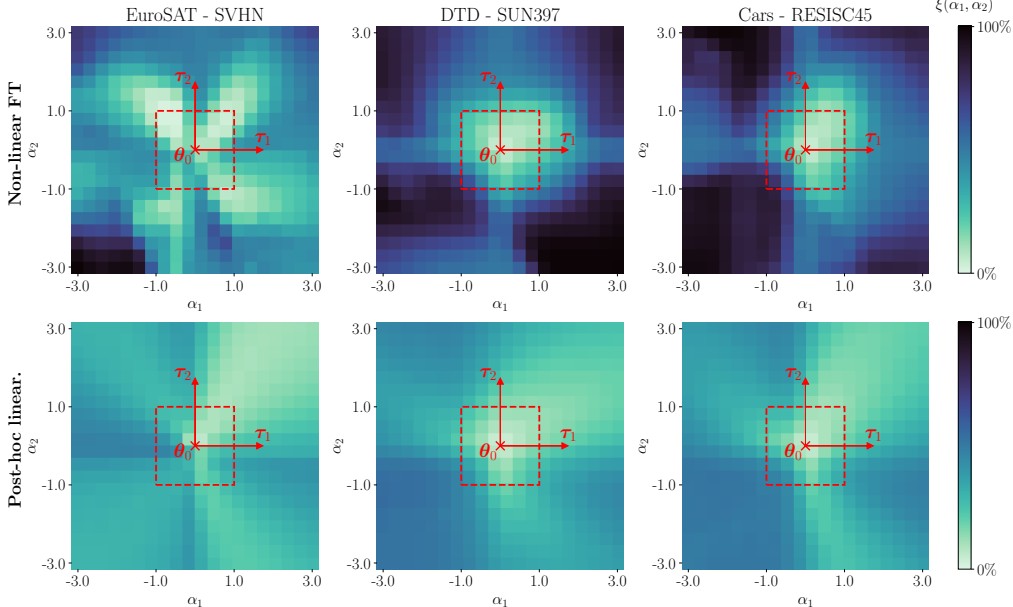

Figure 3: **Visualization of weight disentanglement.** The heatmaps show the disentanglement error $\xi(\alpha_1, \alpha_2)$ of a non-linear CLIP ViT-B/32 (top) and its post-hoc linearization (bottom) on different example task pairs. The light regions denote areas of the weight space where weight disentanglement is stronger. The red box delimits the search space used to compute the best $\alpha$ in all our experiments.

Comparing the disentanglement error of the non-linear models and their post-hoc linearization reveals an interesting finding: linearized models exhibit greater disentanglement than their non-linear counterparts. This is evident from the more extensive regions with low disentanglement errors in Figure 3 (bottom). This explains why the post-hoc linearized models achieve higher normalized accuracies via task addition (cf. Table 1) and manage to forget more through task negation (cf. Table 2). Paradoxically, however, although the greater disentanglement of linearized models allows them to retain more of their relative performance when edited with task arithmetic, they still perform worse in absolute terms due to the great advantage of the non-linear models in single-task accuracy (cf. Figure 2). This suggests that closing the single-task performance gap between linearized and non-linear models could be a way to enhance task arithmetic. We leverage this idea in the next section.

## 5 Enhancing task arithmetic via linearization

We have seen that linearized models are more weight-disentangled than non-linear ones. However, post-hoc linearization degrades single-task performance. We now demonstrate that enforcing models to fine-tune in the tangent space to their pre-trained initialization significantly improves task arithmetic by reducing the single-task accuracy gap.

Specifically, rather than applying the non-linearly fine-tuned task vectors $\tau = \theta^\star - \theta_0$ to $f_{\text{lin}}$, as in Section 3, we propose to directly obtain the task vectors through explicit fine-tuning in the tangent space as illustrated in Figure 4. That is, given a model $f$, we directly fine-tune its linear approximation $f_{\text{lin}}$ around $\theta_0$ [26]. The fine-tuning process can follow the same protocols used before but with the network parameterization dictated by Eq. (3). Due to the

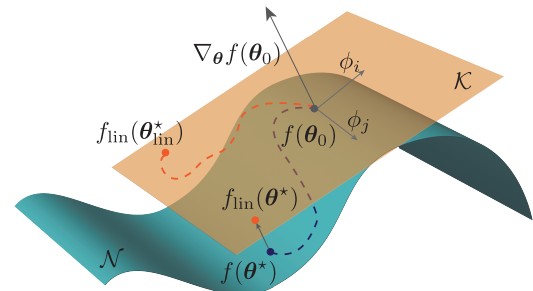

Figure 4: Conceptual illustration of the different approaches we use to edit a pretrained model $f(\,\cdot\,;\boldsymbol{\theta}_0)$. Here $\mathcal{N}$ represents the space of neural network functions $f$, non-linearly parameterized by $\boldsymbol{\theta} \in \boldsymbol{\Theta}$; and $\mathcal{K}$ its tangent space, given by the space of linearized functions $f_{\text{lin}}$.

linear connection between the weight-space and function-space defined in Eq. (3), fine-tuning $f_{\text{lin}}$ is essentially the same as training a kernel predictor with kernel $k_{\text{NTK}}$. As a result, we obtain the fine-tuned weights $\boldsymbol{\theta}_{\text{lin}}^{\star}$ of the linearized model for each task, which allows us to construct the corresponding task vector $\boldsymbol{\tau}_{\text{lin}} = \boldsymbol{\theta}_{\text{lin}}^{\star} - \boldsymbol{\theta}_0$. We provide further details of this procedure in Appendix B.

Moreover, as the considered models do not inherently exhibit linear fine-tuning (see Section 3), this approach yields significantly different results compared to post-hoc linearization, *i.e.*, $f_{\text{lin}}(\boldsymbol{x}; \boldsymbol{\theta}_0 + \boldsymbol{\tau}_{\text{lin}}) \neq f_{\text{lin}}(\boldsymbol{x}; \boldsymbol{\theta}_0 + \boldsymbol{\tau})$. In particular, although both models share the same kernel $k_{\text{NTK}}(\boldsymbol{x}, \boldsymbol{x}')$, the task vectors $\boldsymbol{\tau}_{\text{lin}}$ have been explicitly optimized to maximize the performance of such linearized models. Consequently, by construction, linearized fine-tuning outperforms post-hoc linearization. Indeed, in Figure 5, we observe that linearized fine-tuning significantly reduces the non-linear advantage of non-linear models, as in most cases the performance of $f_{\text{lin}}(\cdot\,; \boldsymbol{\theta}_0 + \boldsymbol{\tau}_{\text{lin}})$ is very similar to the one of $f(\cdot\,; \boldsymbol{\theta}_0 + \boldsymbol{\tau})$ (cf. Figure 2).

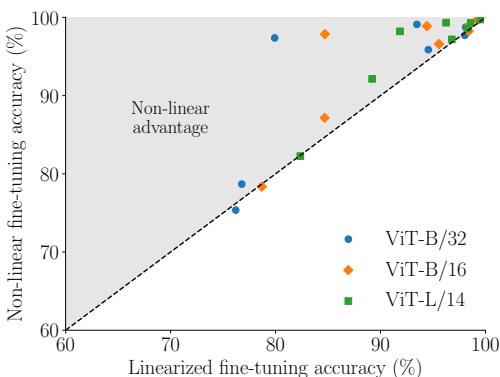

Figure 5: Single-task accuracies of non-linearly FT, $f(\cdot\,; \boldsymbol{\theta}^{\star})$ and linearly FT, $f_{\text{lin}}(\cdot\,; \boldsymbol{\theta}_{\text{lin}}^{\star})$, models.

Remarkably, as we show in Appendix D.4, this increase in single-task performance does not compromise weight disentanglement, which remains as high as for the post-hoc linearized models in Figure 3. As a result, linear fine-tuning allows for improved task arithmetic compared to standard non-linear fine-tuning. In particular, Tables 1 and 2 in their last rows show that linearized fine-tuned models significantly outperform their non-linear counterparts and achieve state-of-the-art results on the task addition and negation benchmarks [39]. The linearized fine-tuned models achieve higher multi-task accuracies through task addition (up to 5.8 points more) and can forget more through task negation (up to 13.1 points more) while maintaining a similar level of accuracy on the control task. Additionally, we observe that the advantage of the linearized models over the non-linear ones is higher for the smaller ViT-B/32 and progressively diminishes as the model size increases up to ViT-L/14[3].

In general, thanks to the efficiency of the Jacobian-vector product implementations in most deep learning frameworks [61], training and inference in linearized neural networks only require an $\mathcal{O}(1)$ increase in computational costs with respect to their non-linear counterparts (see Appendix B for a longer discussion). In this regard, the superiority of task arithmetic of linearized models can make this technique appealing for practical applications. Identifying the right trade-offs between computational cost and performance, as well as faster linearization techniques, is an exciting avenue for future work.

## 6 Towards understanding task arithmetic

We conclude by providing further fundamental insights that can aid our understanding of task arithmetic. In particular, we ask whether any kernel can satisfy Property 1, and we establish a connection between task arithmetic and the spectral properties of the NTK. Then, we argue that weight disentanglement and task arithmetic are emergent properties of pre-training.

### 6.1 Eigenfunction localization

Generally, a kernel $k$ admits a decomposition in terms of a family of eigenfunction-eigenvalue pairs $\{(\phi_\rho, \lambda_\rho)\}_{\rho \in \mathbb{N}}$; which implies that $k$ can only represent functions of the form $f^{\star}(\boldsymbol{x}) = \sum_{\rho=1}^{\infty} c_\rho \phi_\rho(\boldsymbol{x})$ with a finite kernel norm, *i.e.*, $\|f^{\star}\|_{\mathcal{H}}^2 = \sum_{\rho=1}^{\infty} c_\rho^2/\lambda_\rho < +\infty$. Specifically, the coefficients $\{c_\rho\}_{\rho \in \mathbb{N}}$ constitute a representation of the function $f^{\star}$ in the kernel basis.

Consider $T$ tasks $\{f_t^{\star}\}_{t \in [T]}$ supported in their respective non-intersecting domains $\{\mathcal{D}_t\}_{t \in [T]}$. Furthermore, let $\{\phi_\rho\}_{\rho \in \mathbb{N}}$ be an orthogonal basis of eigenfunctions that diagonalizes the kernel on the union of all $\mathcal{D}_t$'s. The following proposition provides a sufficient condition on the representation of the tasks in this basis to ensure the task arithmetic property:

---

[3]In Appendix D.3, we observe that larger models are more weight disentangled.

**Proposition 1** (Simplified). *Suppose that $\{f_t^\star\}_{t \in [T]}$ can be represented by the kernel $k$. The kernel $k$ is capable of performing task arithmetic with respect to $\{f_t^\star\}_{t \in [T]}$ and $\{\mathcal{D}_t\}_{t \in [T]}$ if, for each task $t$, there exists a subset of localized eigenfunctions such that i) $\mathrm{supp}(\phi) \subseteq \mathcal{D}_t$ for each $\phi$ in the subset, and ii) the representation of $f_t^\star$ only involves these basis functions.*

The proof and formal statement are deferred to Appendix C. Intuitively, if each task is represented with eigenfunctions that vanish outside the spatial region identified by the task support, the functions corresponding to different tasks do not interfere. Based on Proposition 1, it is natural to examine whether the NTK of CLIP models displays eigenfunctions localized in each task domain and if it represents the different tasks using these functions. According to the *representer theorem* of kernels [77], after linear fine-tuning on task $t$ with a training set $\{(\boldsymbol{x}_\nu, f_t^\star(\boldsymbol{x}_\nu))\}_{\nu \in [n_t]}$ and $\boldsymbol{x}_\nu \sim \mu_t$, the CLIP's predictor evaluated at a new point $\boldsymbol{x} \in \mathcal{X}$ can be expressed as a linear combination of its kernel $k_{\mathrm{NTK}}$ evaluated on $\boldsymbol{x}$ and the training data, *i.e.*, $f_{\mathrm{lin}}(\boldsymbol{x}) = f(\boldsymbol{x}; \boldsymbol{\theta}_0) + \sum_{\nu \in [n_t]} \beta_\nu \, k_{\mathrm{NTK}}(\boldsymbol{x}_\nu, \boldsymbol{x})$.

To explore whether CLIP models use localized eigenfunctions for task arithmetic, we diagonalize the matrix $(K_{\mathrm{NTK}})_{ij} = k_{\mathrm{NTK}}(\boldsymbol{x}_i, \boldsymbol{x}_j)$ with $\boldsymbol{x}_i \in \mathcal{D}_t$, *i.e.*, the task on which we trained, and $\boldsymbol{x}_j \in \mathcal{D}_t \cup \mathcal{D}_{t'}$, where $\mathcal{D}_{t'}$ is the support of a control task. If the eigenfunctions used to represent $f^\star(\boldsymbol{x})$ are localized, then the power of the eigenvectors of $K_{\mathrm{NTK}}$ must be concentrated in the points belonging to the dataset used for training. To measure this concentration, we introduce the local energy $\mathcal{E}_{\mathrm{loc}}(\boldsymbol{x}) = \sum_\rho \phi_\rho^2(\boldsymbol{x})$, which sums the power of all the eigenfunctions $\phi_\rho$ at a given point $\boldsymbol{x}$.

In Figure 6, we plot this metric for a ViT-B/32 CLIP model trained on RESISC45 with Cars as control. We provide results for other task pairs in Appendix D.8. Notably, the local energy of the eigenfunctions that the predictor uses to represent the RESISC45 task is significantly higher for points belonging to the training dataset. This confirms the presence of eigenfunctions localized across the different data domains and the fact that task arithmetic occurs thanks to the use of those. Indeed, thanks to this localization, CLIP models can effectively separate the representation of different tasks and carry out task-specific operations without interference. We believe that further investigation into this intriguing localization phenomenon holds the potential to deepen our understanding of these models.

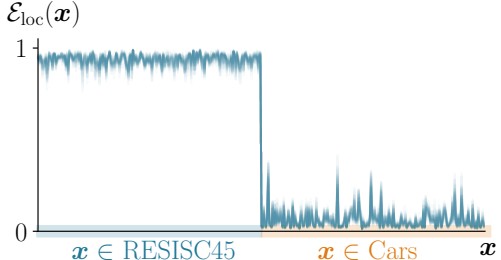

Figure 6: **Eigenfunction localization.** Estimated support of the eigenfunctions of the NTK of a ViT-B/32 CLIP model trained on RESISC45. The plot shows the sum of the local energy of the eigenfunctions over a random subset of the training and control supports (RESISC45 and Cars, respectively).

**Remark.** While we have shown that localized eigenfunctions can play a crucial role in task arithmetic, it is important to note that they are not always necessary. In fact, the task arithmetic property can hold even if the eigenfunctions used to represent a single task cancel outside the corresponding domain. Indeed, although eigenfunctions are linearly independent on the union of the domains, they are not necessarily linearly independent when evaluated on a single domain and, in general, can cancel out. However, if the eigenfunctions maintain their linear independence on each of the domains, *i.e.*, they are *locally* linear independent, then the existence of localized eigenfunctions becomes a necessary condition for task arithmetic. This means that if the eigenfunctions are locally linearly independent and not localized, task arithmetic is not possible. We provide some analytical examples of the latter case in Appendix C, including the NTKs of fully-connected and convolutional networks at initialization.

## 6.2 Weight disentanglement emerges during pre-training

Task arithmetic is not exclusive to CLIP models. In fact, task arithmetic can also be performed on pre-trained text transformers [39, 86], such as GPT-2 [68] or T5 [70] and convolutional neural networks [38] as we also show in Appendices D.5 and D.7. However, it is still unclear if the origin of weight disentanglement comes from pre-training, or if it is a general property of deep networks.

To investigate this, we replicate the task addition experiments but employ randomly initialized ViTs instead of pre-trained ones. The results in Table 3 reveal that task arithmetic is not achievable on randomly initialized ViTs. Indeed, adding task vectors obtained from a random initialization $\boldsymbol{\theta}_0^{\mathrm{rd}}$ does not result in significant improvements in multi-task accuracy over random chance. This

Table 3: **Task addition from random initialization.** We use the same setup as for the experiments in Table 1 but with task vectors obtained from fine-tuning randomly initialized ViTs. Results compare the average single-task accuracy (%) after fine-tuning and the multi-task accuracy (%) via task addition.

| Method | | ViT-B/32 | | ViT-B/16 | | ViT-L/14 | |
|---|---|---|---|---|---|---|---|
| | | Sing. ($\uparrow$) | Multi ($\uparrow$) | Sing. ($\uparrow$) | Multi ($\uparrow$) | Sing. ($\uparrow$) | Multi ($\uparrow$) |
| Random init | $f(\cdot\,;\boldsymbol{\theta}_0^{\mathrm{rd}})$ | 5.3 | – | 4.8 | – | 5.2 | – |
| Non-lin. FT | $f(\cdot\,;\boldsymbol{\theta}_0^{\mathrm{rd}}+\boldsymbol{\tau}^{\mathrm{rd}})$ | 48.5 | 5.5 | 40.6 | 4.5 | 18.0 | 4.8 |
| Linear. FT | $f_{\mathrm{lin}}(\cdot\,;\boldsymbol{\theta}_0^{\mathrm{rd}}+\boldsymbol{\tau}_{\mathrm{lin}}^{\mathrm{rd}})$ | 27.8 | 3.8 | 24.7 | 4.0 | 24.8 | 6.1 |

holds true for both non-linear task vectors, $\boldsymbol{\tau}^{\mathrm{rd}}$, and linearized ones, $\boldsymbol{\tau}_{\mathrm{lin}}^{\mathrm{rd}}$. In Appendix D.9, we further corroborate these findings by computing the disentanglement error and the NTK spectrum of randomly initialized models.

Therefore, we conclude that task arithmetic is a property acquired during pre-training. This observation goes beyond the traditional representation learning view of pre-training, emphasizing that pre-training not only leads to semantically disentangled feature representations but also to the disentanglement of the weights that govern the output on those semantic sets. Investigating the pre-training dynamics that give rise to such disentanglement is another interesting avenue for future research.

# 7 Related work

**Weight interpolation and task arithmetic.**    A growing body of work is exploring the use of interpolations between model weights and task arithmetic to manipulate and enhance the capabilities of pre-trained models. In particular, several studies have shown that interpolating between a model's fine-tuned weights and its pre-trained initialization can lead to improved performance on single tasks, even surpassing their fine-tuning accuracies [27, 40, 57, 71, 72, 87]. In the multi-task setting, averaging the parameters of multiple fine-tuned models has been proposed to produce superior multi-task models [38, 39, 46, 86, 89] that avoid catastrophic forgetting [28, 58] and even provide a better starting point for subsequent fine-tuning [17, 23]. Interestingly, the benefits of weight ensembles and interpolations extend to models trained from scratch, as long as they are properly aligned before merging [3, 79]. This phenomenon has been observed to enhance downstream performance, further emphasizing the potential of weight interpolation and task arithmetic techniques such as the ones studied in this work.

**Linear *vs* non-linear regime.**    Extensive research has been conducted on comparing generalization and dynamical properties of neural networks in linear and non-linear regimes [8, 26, 30, 65, 67, 82] and investigating specific inductive biases [2, 7, 19, 53, 59, 81, 90]. In addition to theoretical understanding, several studies have applied linearized models for practical purposes, such as predicting fine-tuning generalization [22] and training speed [91], as well as enhancing calibration [52] and few-shot performance [5]. Our work serves as another example of the utility of linearized models in certain scenarios where they do not only offer practical benefits but also provide valuable theoretical insights.

**Feature disentanglement.**    The notion of feature disentanglement lies at the heart of representation learning, where ideal representations are assumed to separate distinct data variation factors along different directions in the feature space [1, 10, 35]. A multitude of approaches in generative modeling [14, 34, 73] and self-supervised learning [6, 13, 48, 69] strive to achieve this goal. Our investigation, however, explores a distinct aspect: *weight disentanglement* within the framework of task arithmetic. Departing from the static perspective of feature disentanglement, weight disentanglement connects weight space and function space transitions, thereby enriching our understanding of disentanglement in neural networks from a functional standpoint. Several studies have previously attempted to exploit a similar notion by inducing the learning of task-specific subnetworks within a larger network [32, 36, 54–56, 83, 85]. To the best of our knowledge, our work is the first to demonstrate the natural emergence of such phenomena in specific semantically meaningful tasks during CLIP pre-training.

# 8  Conclusion

In this work, we conducted a thorough analysis of task arithmetic in deep neural networks, delving into its fundamental mechanisms and enhancing its performance. Our findings demonstrate that linearized models, governed by the NTK, outperform their non-linear counterparts in task arithmetic, thus providing a more effective approach for model editing. Crucially, we revealed that weight disentanglement plays a vital role in the success of task arithmetic, as distinct directions in weight space correspond to localized areas in the function space; and that it is an emergent property of pre-training.

A fascinating open question consists in understanding how weight disentanglement arises during pre-training and finding algorithms that enhance it. Another exciting research direction is investigating the potential of tangent spaces for editing other pre-trained models. In this sense, developing more efficient linearized models would be a significant leap forward in this field. These advancements could pave the way for novel approaches to model editing and deepen our understanding of the intricate relationship between weight space and function space in deep learning.

## Acknowledgements

We thank Nikolaos Dimitriadis, Adam Hazimeh, Pau de Jorge, Nikolaos Karalias, Seyed Mohsen Moosavi-Dezfooli, Antonio Sclocchi, Thibault Séjourné, and Matthieu Wyart for helpful feedback and comments. We also thank Gabriel Ilharco for helping set up the code to reproduce their experiments.

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

# A Experimental details

All our experiments were performed using the same hardware consisting of four V100 NVIDIA GPUs with 32GB of memory each and can be reproduced in less than 350 GPU hours. The details of each experiment are the following.

**Fine-tuning.**   All the fine-tuning experiments follow the same training protocol specified in Ilharco et al. [39] with minor modifications to the training code to use linearized models when needed. In particular, we fine-tune all datasets starting from the same CLIP pre-trained checkpoint downloaded from the `open_clip` repository [37]. We fine-tune for $2,000$ iterations with a batch size of $128$, learning rate of $10^{-5}$ and a cosine annealing learning rate schedule with 200 warm-up steps and the AdamW optimizer [49]. As introduced in Ilharco et al. [38], during fine-tuning, we freeze the weights of the classification layer obtained by encoding a standard set of *zero-shot* template prompts for each dataset. Freezing this layer does not harm accuracy and ensures that no additional learnable parameters are introduced during fine-tuning [38]. We use this exact same protocol to fine-tune the non-linear and linearized models and do not perform any form of hyperparameter search in our experiments.

**Tuning of $\alpha$ in task arithmetic benchmarks.**   As in Ilharco et al. [39] we use a single coefficient $\alpha$ to tune the size of the task vectors used to modify the pre-trained models. This is equivalent to setting $\alpha = \alpha_1 = \ldots \alpha_T$ in Eq. (1). Both in the task addition and task negation benchmarks, after fine-tuning, we evaluate different scaling coefficients $\alpha \in \{0.0, 0.05, 0.1, \ldots, 1.0\}$ and choose the value that achieves the highest target metric on a small held-out proportion of the training set as specified in Ilharco et al. [39]. Namely, maximum normalized average accuracy, and minimum target accuracy on each dataset that still retains at least $95\%$ of the accuracy of the pre-trained model on the control task; for task addition and negation, respectively. The tuning of $\alpha$ is done independently for non-linear FT, linearized FT, and post-hoc linearization.

**Normalized accuracies in task addition.**   Table 1 shows the normalized accuracies after editing different models by adding the sum of the task vectors on 8 tasks $\boldsymbol{\tau} = \sum_t \boldsymbol{\tau}_t$. Here, the normalization is performed with respect to the single-task accuracies achieved by the model fine-tuned on each task. Mathematically,

$$\text{Normalized accuracy} = \frac{1}{T} \sum_{t=1}^{T} \frac{\underset{\boldsymbol{x} \sim \mu_t}{\text{acc}} \left[ f(\boldsymbol{x}; \boldsymbol{\theta}_0 + \sum_{t'} \boldsymbol{\tau}_{t'}) \right]}{\underset{\boldsymbol{x} \sim \mu_t}{\text{acc}} \left[ f(\boldsymbol{x}; \boldsymbol{\theta}_0 + \boldsymbol{\tau}_t) \right]}. \tag{7}$$

**Disentanglement error.**   To produce the weight disentanglement visualizations of Figure 3 we compute the value of $\xi(\alpha_1, \alpha_2)$ on a $20 \times 20$ grid of equispaced values in $[-3, 3] \times [-3, 3]$. To estimate the disentanglement error, we use a random subset of $2,048$ test points for each dataset.

**NTK eigenfunction estimation.**   We use the finite-width NTK implementation from the `functorch` sublibrary of PyTorch [66] to compute the $K_{\text{NTK}}$ matrices described in Section 6.1. In particular, we use a random subset of 200 training points for each dataset and compute the singular value decomposition (SVD) of $K_{\text{NTK}}$ to estimate the entries of $\phi_\rho$ on each dataset. As described in Bordelon et al. [11], and to avoid a high memory footprint, we estimate a different set of singular vectors for each output class, equivalent to estimating one kernel matrix per output logit. Figure 6 shows the values of $\mathcal{E}_{\text{loc}}(\boldsymbol{x})$ for each class with a different line. However, there is little variability of the NTK among classes, and hence all curves appear superimposed in the figure.

# B Implementation aspects of linearized models

We now provide more details of the different implementation aspects of linearized models, including basic code and a discussion on their computational complexity.

**Practical implementation.**   Creating linearized models of a neural network is very simple using the `functorch` sublibrary of PyTorch. Specifically, using the fast Jacobian-vector implementation of this library, we can easily create a custom class that takes any `nn.Module` as input and generates a trainable linearized version of it around its initialization. We give a simple example of this in

Listing 1, where we see that the resulting `LinearizedModel` can be directly used in any training script as any other neural network.

In our experiments, we linearize the ViT image encoder of CLIP as the text encoder is frozen in our experiments. In this regard, during training and inference, as it is common in standard CLIP models [69], we normalize the output of the linearized image encoder prior to performing the inner product with the text embeddings. This normalization does not change the classification decision during inference, but it has a rescaling effect on the loss that can influence training dynamics. In our fine-tuning experiments, we found this standard normalization technique has a clearly positive effect in single-task accuracy both for the non-linear and linearized models.

```python
import copy
import torch.nn as nn
from functorch import jvp, make_functional_with_buffers

class LinearizedModel(nn.Module):
    """ Creates a linearized version of any nn.Module.

    The linearized version of a model is a proper PyTorch model and can be
    trained as any other nn.Module.

    Args:
        init_model (nn.Module): The model to linearize. Its parameters are
        used to initialized the linearized model.
    """
    def __init__(self, init_model):
        # Convert models to functional form.
        func, params0, buffers0 = make_functional_with_buffers(init_model)

        # Store parameters and forward function.
        self.func0 = lambda params, x: func(params, buffers0, x)
        self.params0 = params0                    # Initialization parameters.
        self.params = copy.deepcopy(params0)      # Trainable parameters.

        # Freeze initial parameters and unfreeze current parameters.
        for p0 in self.params0: p0.requires_grad = False
        for p in self.params: p.requires_grad = True

    def __call__(self, x):
        # Compute linearized model output.
        dparams = [p - p0 for p, p0 in zip(self.params, self.params0)]
        out, dp = jvp(self.func0, (self.params0,), (dparams,))
        return out + dp
```

Listing 1: Basic PyTorch code to linearize a model.

**Computational complexity.** Jacobian-vector products can be computed efficiently, at the same marginal cost as a forward pass, using forward-mode automatic differentiation rules [9]. This means that doing inference with a linearized model usually takes around two or three times more than with its non-linear counterpart, as for every intermediate operation in the forward pass, its derivative also needs to be computed and evaluated.

Training the linearized models, on the other hand, uses the backpropagation algorithm which, for every forward pass, requires another backward pass to compute the gradients. In this regard, the computational cost of obtaining the gradient with respect to the trainable parameters of the linearized models $\nabla_{\boldsymbol{\theta}} f_{\mathrm{lin}}(\boldsymbol{x}; \boldsymbol{\theta})$ is also roughly twice the cost of obtaining the gradient of its non-

linear counterparts $\nabla_{\boldsymbol{\theta}} f(\boldsymbol{x}; \boldsymbol{\theta})$. Similarly, as the forward-mode differentiation required to compute the forward pass also depends on the values of the derivatives at this step, the final memory footprint of training with the linearized models is also double than the one of training the non-linear ones.

## C  Spectral analysis of linearized models

In this section, we present the formal statement and proof of Proposition 1. Additionally, we delve deeper into the question of whether eigenfunction localization is a necessary condition for task arithmetic and provide analytical examples with exactly diagonalizable NTKs to support our discussion.

**Proposition 2** (Formal version of Proposition 1). *Suppose that the task functions $\{f_t^\star\}_{t\in[T]}$ belong to the RKHS of the kernel $k$ and their coefficients in the kernel eigenbasis are $\{(c_{t,\rho}^\star)_{\rho\in\mathbb{N}}\}_{t\in[T]}$. If $\forall t, \rho$, either $c_{t,\rho}^\star = 0$ or $\mathrm{supp}(\phi_\rho) \subseteq \mathcal{D}_t$, then the kernel $k$ has the task arithmetic property with respect to $\{f_t^\star\}_{t\in[T]}$ and $\{\mathcal{D}_t\}_{t\in[T]}$ .*

*Proof.* The task arithmetic property requires that $\forall t' \in [T]$, $\forall \boldsymbol{x} \in \mathcal{D}_{t'}$, $\sum_{t\in[T]} f_t^\star(\boldsymbol{x}) = f_{t'}^\star(\boldsymbol{x})$. Representing the task functions in the kernel basis, we have

$$\forall t' \in [T], \ \forall \boldsymbol{x} \in \mathcal{D}_{t'}, \ \sum_{t\in[T]} \sum_{\rho\in\mathbb{N}} c_{t,\rho}^\star \phi_\rho(\boldsymbol{x}) = \sum_{\rho\in\mathbb{N}} c_{t',\rho}^\star \phi_\rho(\boldsymbol{x}). \tag{8}$$

This condition can be rewritten as

$$\int_{\mathcal{D}_{t'}} \left( \sum_{t\in[T], \, t\neq t'} \sum_{\rho\in\mathbb{N}} c_{t,\rho}^\star \phi_\rho(\boldsymbol{x}) \right)^2 d\boldsymbol{x} = 0. \tag{9}$$

If, for each $t$, the eigenfunctions corresponding to non-zero coefficients are supported within a subset of $\mathcal{D}_t$ and all domains $\mathcal{D}_t$'s are disjoint, then all the summands inside the integral in Eq. (9) become zero inside $\mathcal{D}_{t'}$, and thus the proof is complete. $\square$

As we discussed in Section 6.1, eigenfunction localization is generally not a necessary condition to achieve task arithmetic. However, we now show that if the eigenfunctions are locally linear independent across the different task domains, then the localization property becomes a necessary condition for task arithmetic. The proposition presented below formalizes this concept.

**Proposition 3.** *Suppose that the task functions $\{f_t^\star\}_{t\in[T]}$ belong to the RKHS of the kernel $k$ and their coefficients in the kernel eigenbasis are $\{(c_{t,\rho}^\star)_{\rho\in\mathbb{N}}\}_{t\in[T]}$. Furthermore, let the kernel eigenfunctions be either zero or linearly independent over each domain $\mathcal{D}_t$. The kernel $k$ has the task arithmetic property with respect to $\{f_t^\star\}_{t\in[T]}$ and $\{\mathcal{D}_t\}_{t\in[T]}$ if and only if $\forall t, \rho$, either $c_{t,\rho}^\star = 0$ or $\mathrm{supp}(\phi_\rho) \subseteq \mathcal{D}_t$.*

*Proof.* The initial steps of the proofs follow those of the previous proposition. In particular, let's consider the integral in Eq. (9). Due to the linear independence of the non-zero kernel eigenfunctions on $\mathcal{D}_{t'}$, for this integral to be zero, we have only two possibilities: either *i)* all coefficients $\{(c_{t,\rho}^\star)_{\rho\in\mathbb{N}}\}_{t\in[T], \, t\neq t'}$ must be zero or *ii)* the eigenfunctions corresponding to non-zero coefficient $c_{t,\rho}^\star$ $(t \neq t')$ must be zero in $\mathcal{D}_{t'}$. Since the proposition is valid for any set of functions, condition *i)* is not feasible. Therefore, condition *ii)* must hold. Furthermore, since Eq. (9) is valid $\forall t' \in [T]$, it follows that the eigenfunctions used to represent each task $t'$ are zero in $\overline{\mathcal{D}}_{t'} = \bigcup_{t\in[T], \, t\neq t'} \mathcal{D}_t$. Consequently, these eigenfunctions are only supported in $\mathcal{D}_{t'}$ or a subset thereof. $\square$

In order to understand the implications of this proposition, it is useful to examine simple data geometries and architectures for which the NTK can be analytically diagonalized. For instance, when data is uniformly distributed on a ring or a torus, the NTK of fully-connected and convolutional neural networks at initialization can be diagonalized with the Fourier series [12, 25, 29, 75]. Fourier atoms are linearly independent on any interval [18] and not localized. Consequently, according to Proposition 3, these architectures cannot perform task arithmetic within such settings. This straightforward calculation aligns with the observation that task arithmetic generally emerges as a property of pre-training and is not inherently present at initialization, as we numerically demonstrated for CLIP models in Section 6.2.

# D  Further experimental results

We now present additional experiments that expand the findings discussed in the main text.

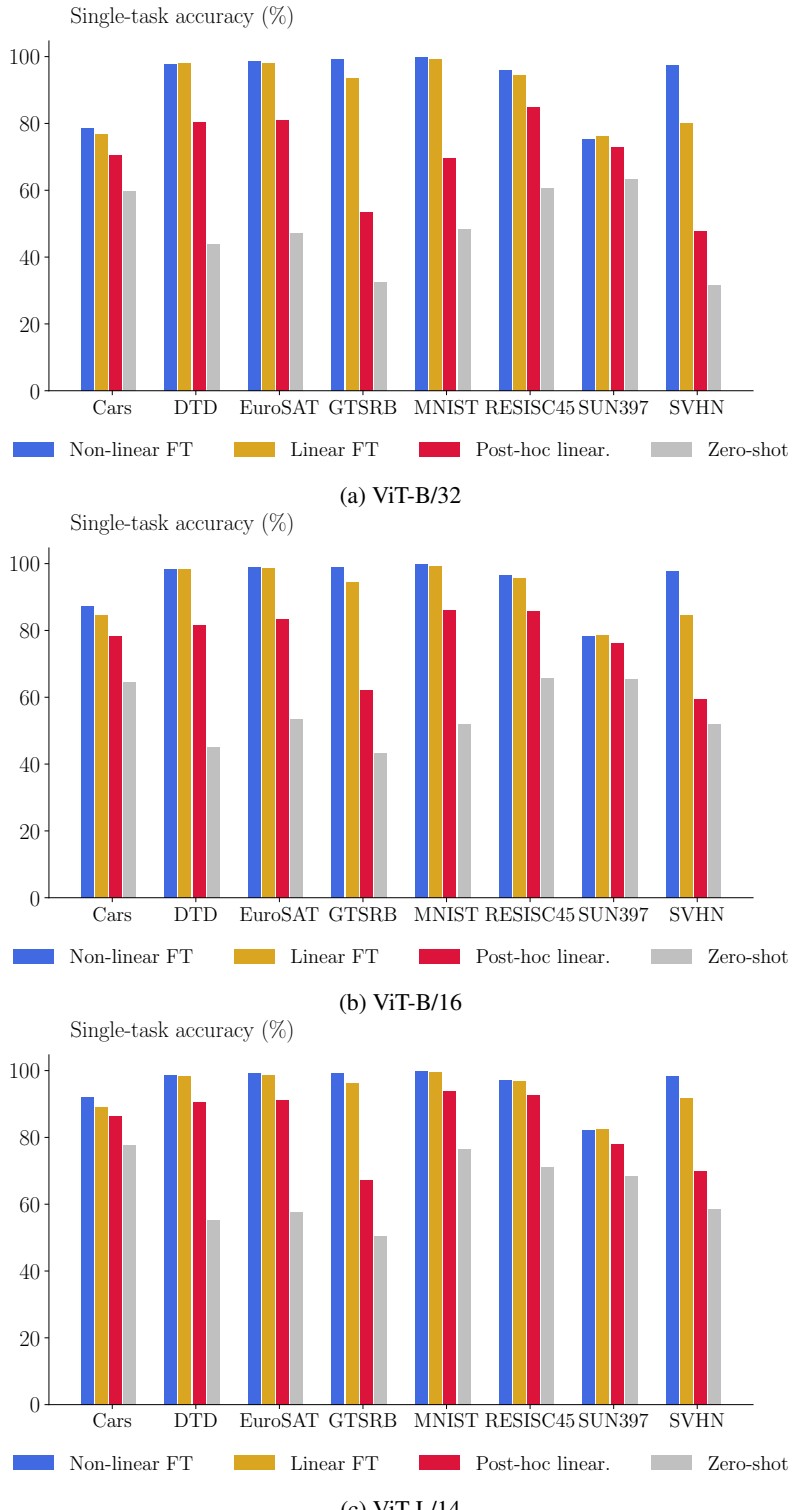

Figure 7: **Single-task accuracies (CLIP).** Accuracy of different models obtained using different strategies on each of the tasks.

### D.1 Fine-tuning accuracies

In Figure 7, we report the single-task accuracies achieved by different CLIP models before fine-tuning (referred to as *zero-shot*), after fine-tuning with different dynamics (referred to as *non-linear FT* and *linear FT*), and after linearizing the non-linearly fine-tuned models (*post-hoc linearization*).

These results demonstrate that non-linear fine-tuning consistently achieves the highest accuracy, indicating a *non-linear advantage*. However, an interesting observation is that the gap between non-linear, linear, and post-hoc linearized models diminishes as the model size increases. This trend can be explained by the fact that larger models, which are more over-parameterized, inherently induce a stronger kernel behavior during fine-tuning. As a result. they tend to stay closer to the NTK approximation, closing the gap with linearized models.

### D.2 Detailed results on task addition

In addition to the results presented in Table 1 in the main text, we report in Figure 8 the absolute accuracies of different CLIP models on the single tasks before (*zero-shot*) and after performing task addition with different strategies (*non-linear fine-tuning*, *post-hoc linearization*, and *linear fine-tuning*).

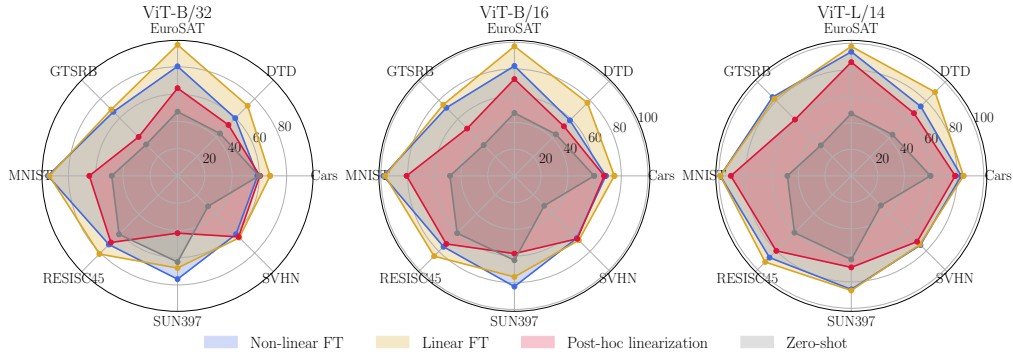

Figure 8: **Task addition performance.** Absolute accuracy (%) of each task after performing task addition with different linear/non-linear strategies and over different CLIP ViT models.

For all models and all datasets, except SUN397 [88], we observe that linearized task arithmetic achieves the highest accuracies. Interestingly, as commented in the main text, the gap in performance between linear and non-linear task vectors decreases while increasing the size of the model. This observation aligns with the previous observation that fine-tuning with larger models is better approximated by the NTK description.

## D.3 Weight disentanglement and model scale

We note that weight disentanglement can also explain the increased performance of task arithmetic with model scale. As we see in Figure 9, the size of the areas with a low disentanglement error grows with the model scale. One plausible explanation for this is that larger models inherently induce a stronger kernel behavior during fine-tuning. Namely, since the models have more parameters, each parameter has to change less to fit the training examples. As a result, they tend to stay closer to the NTK approximation, closing the gap with linearized models and taking benefit of the better weight disentanglement of the models lying in the tangent space.

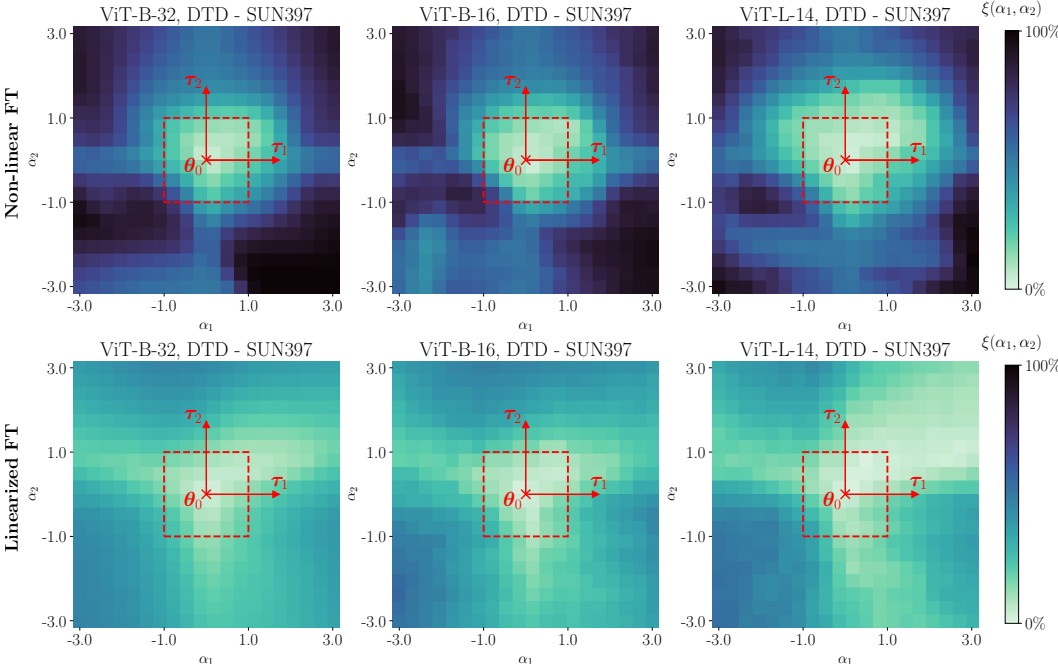

Figure 9: **Weight disentanglement and model scale.** The heatmaps show the disentanglement error $\xi(\alpha_1, \alpha_2)$ of different non-linear CLIP ViTs (top) and their post-hoc linearizations (bottom) on DTD and SUN397. The light regions denote areas of the weight space where weight disentanglement is stronger. The red box delimits the search space used to compute the best $\alpha$ in all our experiments.

### D.4 Weight disentanglement of linearized and random models

In Figure 10, we present the disentanglement error of a linearized CLIP ViT-B/32 model across three different dataset pairs. By comparing these results with Figure 3, we can clearly observe that linearized models exhibit significantly more weight disentanglement compared to their non-linear counterparts, similar to the findings obtained for post-hoc linearization.

Conversely, in Figure 11, we showcase the disentanglement error of a CLIP ViT-B/32 model that was non-linearly fine-tuned starting from a random initialization. In all panels, we observe a high disentanglement error, which supports the claim that weight disentanglement and, consequently, task arithmetic are emergent properties of pre-training.

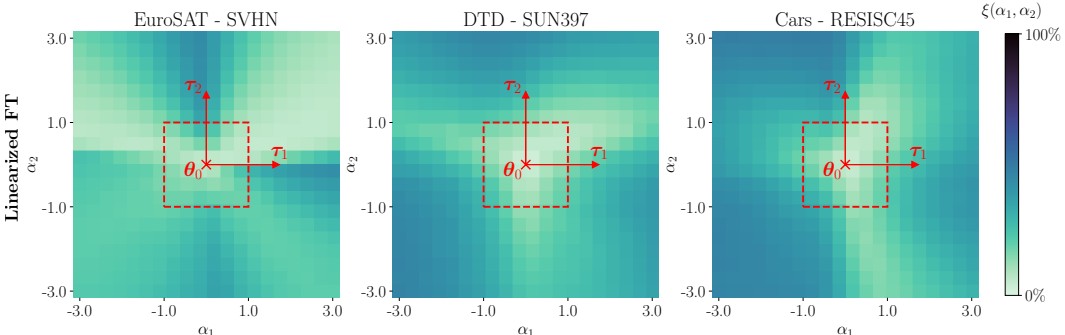

Figure 10: **Visualization of weight disentanglement from linearized models.** The heatmaps show the disentanglement error $\xi(\alpha_1, \alpha_2)$ of a ViT-B/32 linearly fine-tuned on different example task pairs. The light regions denote areas of the weight space where weight disentanglement is stronger. The red box delimits the search space used to compute $\alpha$ in our experiments.

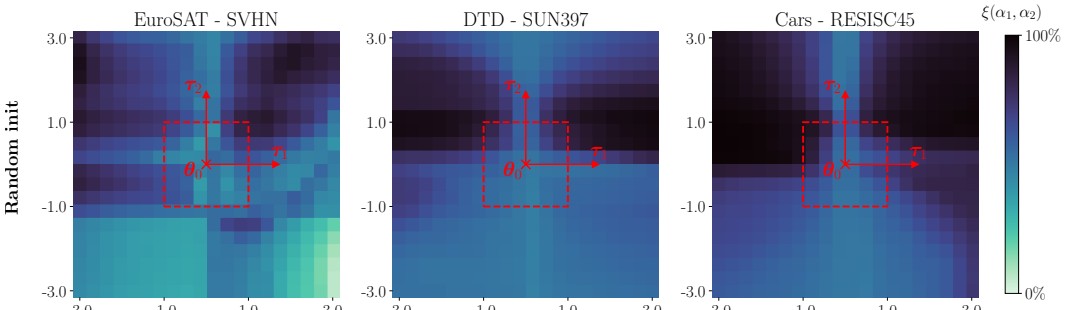

Figure 11: **Visualization of weight disentanglement from random initialization.** The heatmaps show the disentanglement error $\xi(\alpha_1, \alpha_2)$ of a ViT-B/32 fine-tuned from a random initialization on different example task pairs. The light regions denote areas of the weight space where weight disentanglement is stronger. The red box delimits the search space used to compute $\alpha$ in our experiments.

## D.5 Task arithmetic with a convolutional architecture

We replicate our task addition experiments using a convolutional architecture rather than a ViT. Specifically, we finetune a ConvNeXt [47] pre-trained on LAION-400M using CLIP [78] on the 8 tasks from our task addition benchmark. We observe that also for this architecture linearized fine-tuning improves task arithmetic performance (see Table 4).

Table 4: **Task addition with a CNN.** Average absolute (%) and normalized accuracies (%) of a CLIP ConvNeXt edited by adding the sum of the task vectors of 8 tasks. We report results for the non-linear and linearized models normalizing performance by their single-task accuracies.

| Method | | ConvNeXt | |
| --- | --- | --- | --- |
| | | Abs. ($\uparrow$) | Norm. ($\uparrow$) |
| Pre-trained | $f(\boldsymbol{\theta}_0^{\mathrm{rd}})$ | 57.5 | – |
| Non-lin. FT | $f(\boldsymbol{\theta}_0^{\mathrm{rd}} + \boldsymbol{\tau}^{\mathrm{rd}})$ | 79.1 | 83.6 |
| Linear. FT | $f_{lin}(\boldsymbol{\theta}_0^{\mathrm{rd}} + \boldsymbol{\tau}_{lin}^{\mathrm{rd}})$ | **81.1** | **85.7** |

## D.6 Task arithmetic with closed vocabulary models

We also replicate our task addition experiments using a ViT-B/16 pre-trained on ImageNet-1k using standard supervised learning. Note that this is a closed vocabulary model, and therefore, when fine-tuning on the different tasks we need to also fine-tune a randomly initialized head. When interpolating the different task vectors we freeze the resulting head to arrive at the final models.

Interestingly, we find that this closed-vocabulary model can do task arithmetic, albeit at a lower performance (both in terms of absolute and normalized accuracy) than the open vocabulary ones (see Table 5). Remarkably, linearized fine-tuning also enhances disentanglement in this model – it yields higher normalized accuracies – but does not perform better than non-linear task addition due to its lower single-task performance.

Table 5: **Task addition with a closed vocabulary model.** Average absolute (%) and normalized accuracies (%) of a ViT-B/16 pre-trained on ImageNet-1k edited by adding the sum of the task vectors of 8 tasks. We report results for the non-linear and linearized models normalizing performance by their single-task accuracies.

| Method | | ViT-B/16 (supervised) | |
| --- | --- | --- | --- |
| | | Abs. ($\uparrow$) | Norm. ($\uparrow$) |
| Pre-trained | $f(\boldsymbol{\theta}_0^{\mathrm{rd}})$ | 2.3 | – |
| Non-lin. FT | $f(\boldsymbol{\theta}_0^{\mathrm{rd}} + \boldsymbol{\tau}^{\mathrm{rd}})$ | **54.9** | 52.2 |
| Linear. FT | $f_{lin}(\boldsymbol{\theta}_0^{\mathrm{rd}} + \boldsymbol{\tau}_{lin}^{\mathrm{rd}})$ | 41.8 | **66.1** |

## D.7 Weight disentanglement in other architectures and modalities

To substantiate the generality of weight disentanglement, we conduct a new experiment on a pre-trained T5-Base model [70] from Hugging Face Hub [84], fine-tuned on two benchmark Natural Language Processing tasks, i.e., sentiment analysis on movie reviews with the IMDB dataset [51] and question answering with the QASC dataset [42]. The results, illustrated in the right panel in Figure 12, show a notable region around the pre-trained checkpoint characterized by low disentanglement error. This finding echoes the ability of T5 to perform task arithmetic as demonstrated in Ilharco et al. [39], thereby reinforcing the robustness of our conclusions.

Similarly, in Figure 12 we also see that the tangent space of CLIP ConvNeXt models is also more disentangled than the non-linear function space around the pre-trained initialization. The same findings also apply to closed vocabulary models, as shown in Figure 13.

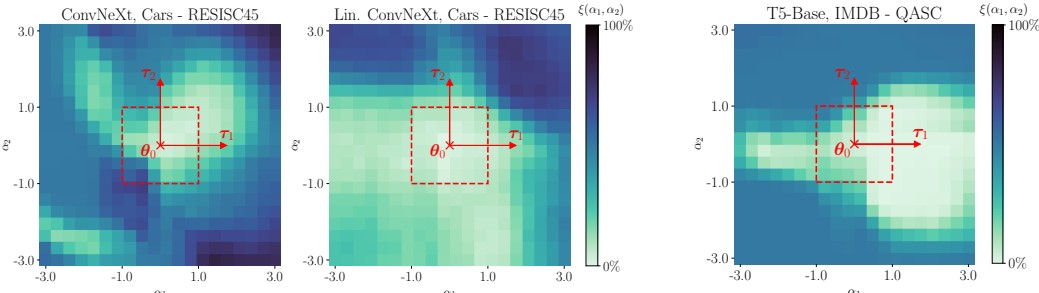

Figure 12: **Visualization of weight disentanglement for other architectures and modalities.** The heatmaps show the disentanglement $\xi(\alpha_1, \alpha_2)$ of a non-linearly and linearly fine-tuned ConvNeXt on a pair of vision tasks (two left panels) and a T5-Base model fine-tuned on a pair of NLP tasks (right).

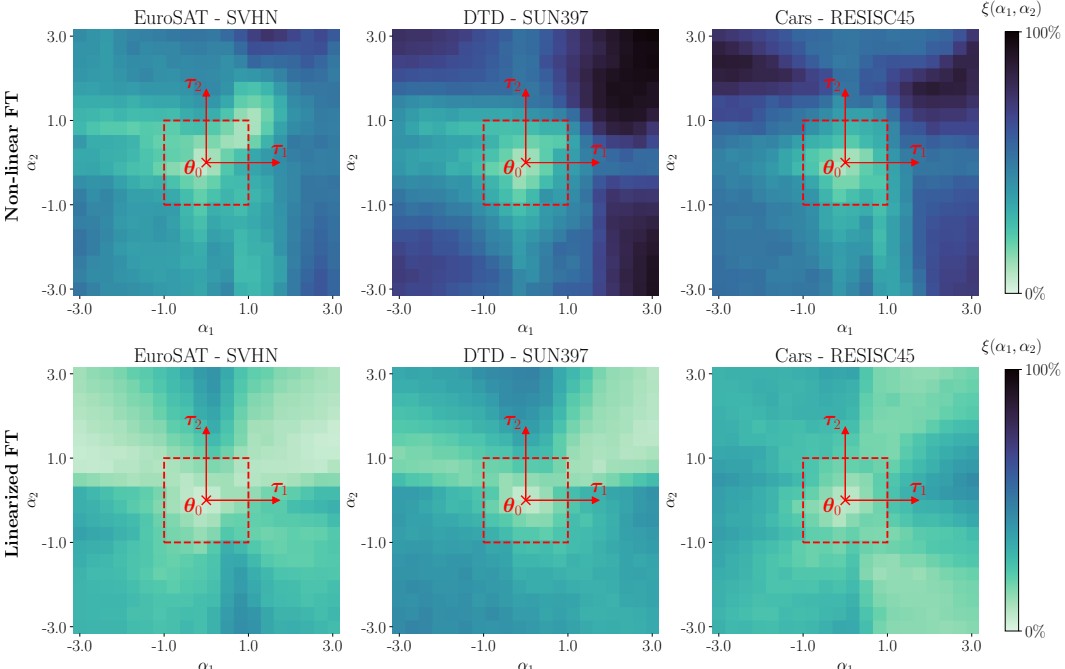

Figure 13: **Weight disentanglement of closed vocabulary models.** The heatmaps show the disentanglement error $\xi(\alpha_1, \alpha_2)$ of a ViT-B/16 model pre-trained on ImageNet-1k using supervised training in the non-linear function space (top) and its linearizations (bottom) on different task pairs. The light regions denote areas of the weight space where weight disentanglement is stronger. The red box delimits the search space used to compute the best $\alpha$ in all our experiments.

## D.8 Localization of eigenfunctions of CLIP's NTK

In Figure 14, we plot the local energy of the NTK eigenfunctions for a pre-trained CLIP ViT-B/32 model evaluated on three different data supports and control data supports. These panels complement the information presented in Figure 6 in the main text, where we observed that the CLIP has eigenfunctions whose energy is concentrated on points belonging to the respective dataset.

In Figure 15, we extend this analysis to a randomly-initialized CLIP ViT-B/32 model. In all panels, we observe a non-trivial but considerably poor degree of eigenfunction localization. This observation aligns with the finding that randomly-initialized linearized models cannot perform task arithmetic. Indeed, as we showed in the previous subsection, in this case the model's weights are not effectively disentangled, hindering its ability to perform task arithmetic operations. In summary, eigenfunction localization offers a complementary perspective on the limitations of randomly-initialized models.

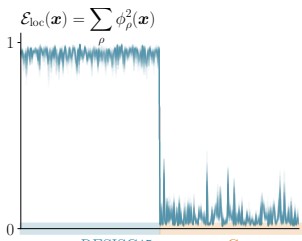 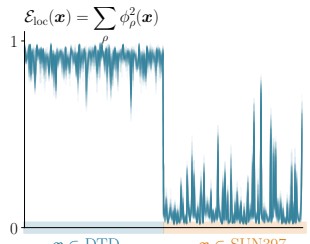 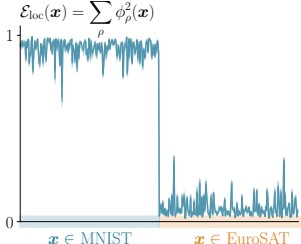

Figure 14: **Eigenfunction localization.** Estimated support of the eigenfunctions of the NTK of a ViT-B/32 CLIP model trained on different datasets. The plot shows the sum of the local energy of the eigenfunctions over a random subset of the training and control supports

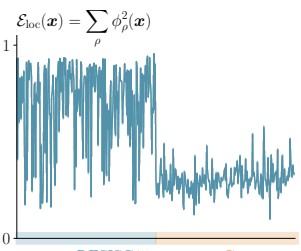 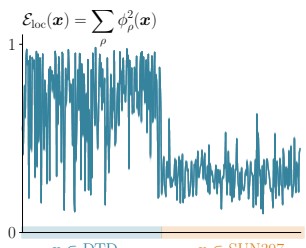 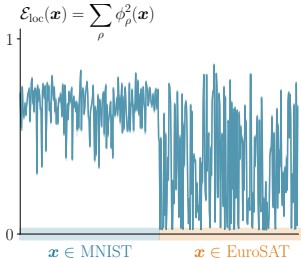

Figure 15: **Eigenfunction localization.** Estimated support of the eigenfunctions of the NTK of a randomly initialized ViT-B/32 model trained on different datasets. The plot shows the sum of the local energy of the eigenfunctions over a random subset of the training and control supports

## D.9   Further experiments with randomly-initialized networks

We conclude by showing, in Figure 16, the absolute single-task accuracy achieved by different CLIP ViT models that were fine-tuned from a random initialization. Both the base models achieve non-trivial or moderate accuracy on the majority of benchmark tasks, using both non-linear and linearized fine-tuning dynamics.

These findings reinforce the intuition that non-pretrained models are not failing in task arithmetic due to their inability to learn the task initially. Instead, as argued earlier, the primary reason for the failure of non-pre-trained models in task arithmetic is their lack of weight disentanglement.

Interestingly, the performance of the randomly-initialized large model is generally poorer compared to the base models. This observation can be attributed to the models' tendency to overfit the training data, which is more likely to occur when a model has a larger capacity.

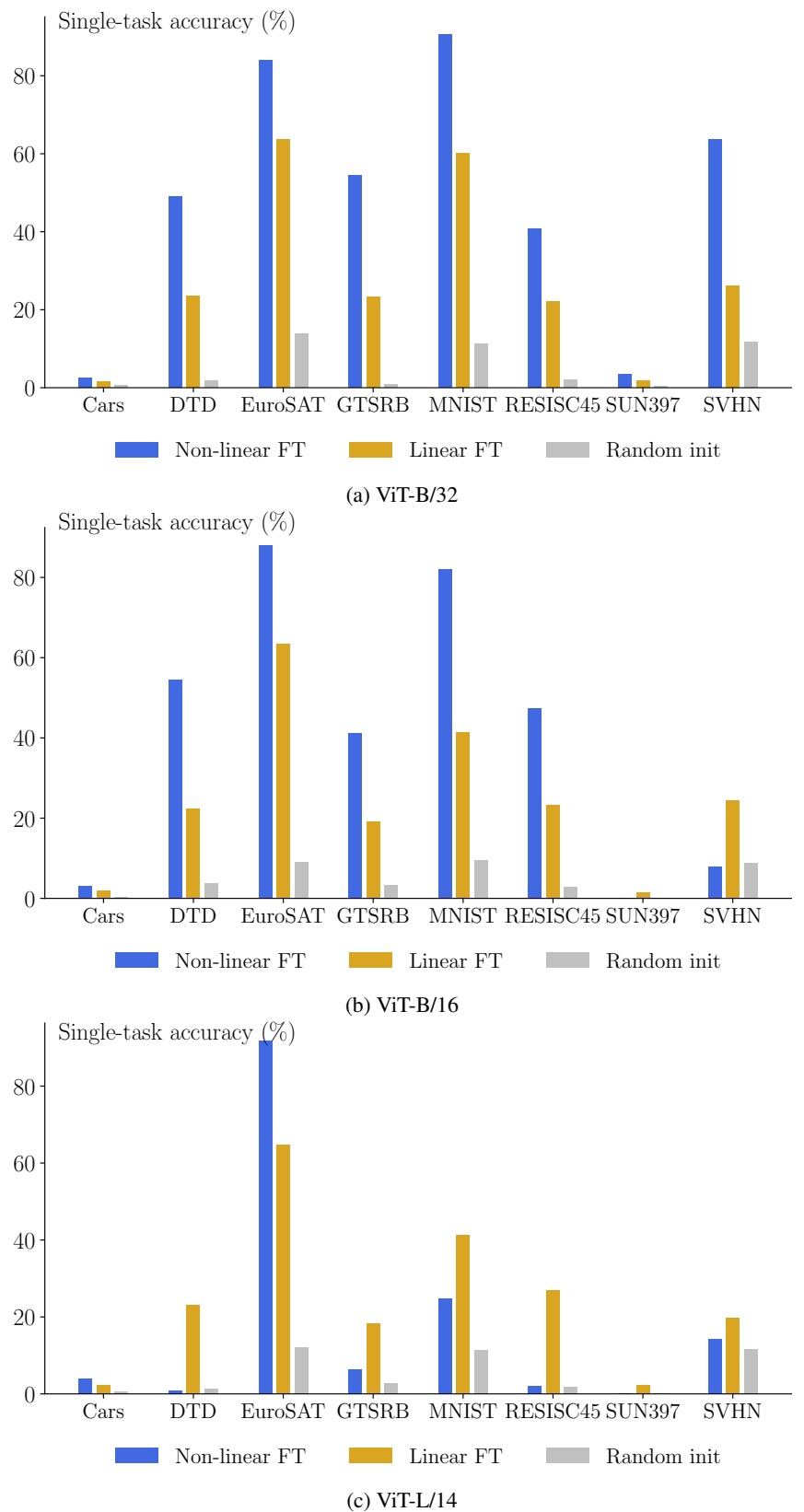

Figure 16: **Single-task accuracies (random init).** Accuracy of different models obtained using different strategies on each of the tasks.

