# OpenReview forum: "Task Arithmetic in the Tangent Space: Improved Editing of Pre-Trained Models"
_NeurIPS.cc/2023/Conference — NeurIPS 2023 oral_

### Official Review · Reviewer_eVrq · 2023-06-09

**Soundness:** 3 good
**Presentation:** 3 good
**Contribution:** 3 good
**Rating:** 6
**Confidence:** 3

**Summary:**

This paper studies the "task vectors" framework where the weights of models can be perturbed in specified directions corresponding to tasks which result in improvements on those tasks. They attribute the success of this framework to "weight disentanglement" which means that adding a task vector for task i does not change how the network behaves when seeing input for task j != i. In addition, their investigation reveals that task vectors are not a consequence of fine-tuning occurring in the NTK regime (post-hoc linearization underperforms), but explicitly linearizing fine-tuning improves the task vectors framework as it "amplifies" "weight disentanglement".

**Strengths:**

Overall I thought this paper was very interesting, it's claims are mostly precise and thoroughly investigated, and it's results are impressive (+ 5.8 for task addition). I though Figure 5 was particularly nice evidence of the authors hypothesis.

**Weaknesses:**

- Limited experiments for many tasks. I would be very curious to see what happens in e.g. table 1 if you scale up the number of tasks.
- Limited attempt to falsify the hypothesis -- for instance can you consider two tasks which share the same input images? Would task vectors still work? (See question in questions section).
- Throughout the paper the authors mention variations of "Specifically, we probe the hypothesis presented in Wortsman et al. [80] that task arithmetic is possible thanks to the fact that models inherently operate in a linear regime" but I cannot actually find anywhere in [80] where this hypothesis is stated? I thought that Wortsman et al. [79, 80] were observing that ensembles behaved roughly similar to averages in the fine-tuning regime, and simply used the NTK regime as an example of where this would occur, but noted that there was still differences between the two (which is not predicted by the NTK regime). How do the authors feel about the hypothesis that averaging models (~= task vectors) ~= output-ensembling + terms-which-may-be-small (e.g., Section 4 of [79]).

**Questions:**

- Do the authors findings point to any "better" way of adding task vectors rather than just adding them for all weights in the network?
- How does weight disentanglement change with scale?
- Can the authors think of an experiment which might falsify their hypothesis? E.g., do they think that task vectors might still "work" if two tasks share the same input space. Here's a quick thought: get one task vector which corresponds to the problem "first 5 classes in CIFAR10 or second 5 classes" and another corresponding to the task "traditional CIFAR10". If you apply both these task vectors then look at CIFAR10 performance on just the first 5 classes I think performance would still be as good as just applying the CIFAR10 task vector?
- Should the community switch to linearized fine-tuning?

**Limitations:**

Would have been nice to see a limitations section.

---

> ### Author Rebuttal · Authors · 2023-08-10
>
> We appreciate that the reviewer reported that our paper is very interesting and has impressive results, and their engagement to improve it. Below, we address their comments.
>
> **Adding more tasks**
>
> For the paper, we are using the experimental setting of Ilharco et al. [39], where task arithmetic was introduced. Namely, our task-addition experiments already involve a set of 8 different vision tasks. Yet, while the scalability of task arithmetic across a larger number of tasks is an appealing question, it goes beyond the paper’s objective and is left for future investigations. In the revised version of our manuscript, we will also include results on NLP tasks and using other convolutional architectures.
>
> **Disjoint task support hypothesis**
>
> Our theoretical analysis specifically focuses on non-overlapping scenarios, consistent with the case which is most studied in the empirical literature. In fact, the hypothesis of having disjoint task supports is satisfied in both Ilharco et al. [39] and our work. Encompassing overlapping task supports – which might require a slight change in the definition of weight disentanglement – remains an interesting problem. However, we do not see studying only the non-overlapping case as a weakness. Indeed, in the context of vision-language tasks, the input space is the Cartesian product of all images and captions, which is a high-dimensional space in which the overlap between tasks might be minimal. We will ensure to add this justification in the revised version.
>
> **Previous linearity hypothesis**
>
> Wortsman et al. [80] observe in Appendix F that even though averaging weights and averaging output functions in practice are not exactly the same, these two methods are not as different as they appear. In particular, they report that these methods are equivalent in the NTK regime and this regime might be an accurate approximation of fine-tuning. Similarly, in Appendix A of Ilharco et al. [39], the authors justify their results on task arithmetic based on the NTK hypothesis for which they also cite Wortsman et al. [80], sharing many of the same authors.
>
> Concerning the question on the hypothesis introduced in [79], we do not have arguments to refute it. In fact, we believe that weight disentanglement might play a role in this setting as well if different models are specializing to different regions of the input space. In that case, the terms-which-may-be-small would be the disentanglement error.
>
> **Effect of scale on weight disentanglement**
>
> Our results reveal that by scaling the number of model parameters, the performance of linearized fine-tuning becomes closer to the one of standard non-linear fine-tuning. As commented briefly in Appendix D.1, one plausible interpretation is that larger models, which are more over-parameterized, inherently induce a stronger kernel behavior during fine-tuning. Namely, since the models have more parameters, each parameter has to change less to fit the training examples. As a result, they tend to stay closer to the NTK approximation, closing the gap with linearized models and taking benefit of the better weight disentanglement of the models lying in the tangent space.
>
> As also suggested by Reviewer [DjkZ](https://openreview.net/forum?id=0A9f2jZDGW&noteId=v6Lz3dFhri), we conducted supplementary experiments that visualize how weight disentanglement varies changing the model scale (see Figure R.2 of the *Author Response document* attached [here](https://openreview.net/forum?id=0A9f2jZDGW&noteId=7E6o5YEkJw)). Consistently to our results, larger models exhibit stronger weight disentanglement, as highlighted by the larger light region in the right panel of the first row of Figure R.2. Yet, interestingly, the models linearly fine-tuned are always more weight disentangled than the non-linearly fine-tuned ones, highlighting the strength of linearized models for model editing. We will add this discussion and the new results to the paper.
>
> **Adoption of linearized fine-tuning**
>
> Linearized fine-tuning, as shown in our paper, gives multiple benefits for, e.g., ensembling, composition, and forgetting (negation). Notably, for convex losses, this method also enjoys a convex optimization landscape and could be used to provide further theoretical guarantees in the future. However, we want to remark that studying the performance of linear vs non-linear models is an ongoing line of research (see, e.g., references in Related Works – Linear vs non-linear regime). In fact, linearized fine-tuning may not be uniformly superior in all settings and architectures, although it seems to be the case for the models we studied in the present work. All in all, we hope that our findings will motivate further empirical exploration to discern in which cases linearized fine-tuning competes effectively with standard non-linear fine-tuning.
>
> **Limitations**
>
> Our limitations are clearly reported in the paper. In particular, we had a section in the Appendix to the O(1) computational complexity increase of linearized models and discussed the limitations of Proposition 1 as a remark in the main body of the text. Following the reviewer’s suggestion, we will emphasize those points further in the revised version of our work.
>
> We thank the reviewer for their valuable feedback and remain available to answer further questions or provide more clarifications regarding the previous points.

---

### Official Review · Reviewer_9cnX · 2023-06-26

**Soundness:** 4 excellent
**Presentation:** 3 good
**Contribution:** 4 excellent
**Rating:** 8
**Confidence:** 4

**Summary:**

This paper presents a comprehensive theoretical and empirical analysis of task arithmetic for model editing, where adding different task vectors (obtained by taking the difference between fine-tuned and pretrained model checkpoints) could improve the model’s performance on these tasks and vice versa. The authors propose weight disentanglement to investigate the underlying principles of task arithmetic, which involves decomposing the learned model function into a sum of localized components with disjoint supports. Specifically, the authors compare **regular** and **post-hoc linearized** fine-tuned models and find that weight disentanglement significantly contributes to the ability of task arithmetic. Based on these insights, the authors then propose to directly employ the linearized model for fine-tuning, obtain optimized task vectors, and lead to improved task arithmetic performance. Further analyses are conducted to reveal the connection between task arithmetic and weight disentanglement.

**Strengths:**

- The paper is clearly written and well organized.
- This work presents a neat analysis of task arithmetic based on the use of linearization and neural tangent kernels, which is novel to my knowledge and interesting, offering a fresh perspective on understanding the geometry of pre-trained checkpoints’ weights.
- The proposed method to further improve task arithmetic is simple yet effective, supported by extensive empirical findings.

**Weaknesses:**

As the authors already mentioned, one potential weakness is the introduced computational overhead during training. However, since the main focus of this work is a comprehensive analysis of task arithmetic, this is not a significant issue in the context of this work.

**Questions:**

1. How does model scale impact weight disentanglement, and thus the benefits of linearized fine-tuning?
2. (Minor) Can the findings about task arithmetic be generalized to natural language tasks as well? By intuition, it seems that the setting of natural language texts is more straightforward for weight disentanglement due to the inherent discreteness of text tokens.

**Limitations:**

The authors have adequately addressed the limitations.

---

> ### Author Rebuttal · Authors · 2023-08-10
>
> We sincerely appreciate the reviewer’s recognition of our work and their engagement to improve it!  Below, we address their comments.
>
> **Effect of scale on weight disentanglement**
>
> Our results reveal that by scaling the number of model parameters, the performance of linearized fine-tuning becomes closer to the one of standard non-linear fine-tuning. As commented briefly in Appendix D.1, one plausible interpretation is that larger models, which are more over-parameterized, inherently induce a stronger kernel behavior during fine-tuning. Namely, since the models have more parameters, each parameter has to change less to fit the training examples. As a result, they tend to stay closer to the NTK approximation, closing the gap with linearized models and taking benefit of the better weight disentanglement of the models lying in the tangent space.
>
> As also suggested by Reviewer [DjkZ](https://openreview.net/forum?id=0A9f2jZDGW&noteId=v6Lz3dFhri), we conducted supplementary experiments that visualize how weight disentanglement varies changing the model scale (see Figure R.2 of the *Author Response document* attached [here](https://openreview.net/forum?id=0A9f2jZDGW&noteId=7E6o5YEkJw)). Consistently to our results, larger models exhibit stronger weight disentanglement, as highlighted by the larger light region in the right panel of the first row of Figure R.2. Yet, interestingly, the models linearly fine-tuned are always more weight disentangled than the non-linearly fine-tuned ones, highlighting the strength of linearized models for model editing. We will add this discussion and the new results to the paper.
>
> **Generalization to NLP tasks**
>
> Task arithmetic and our findings on weight disentanglement can be readily generalized to NLP tasks. In order to show the generality of weight disentanglement, we conducted a new experiment on a pre-trained T5 base model from Hugging Face, fine-tuned on two benchmark NLP tasks (sentiment analysis on movie reviews and question answering). The results, illustrated in the right panel in Figure R.1 of the *Author Response document*, show a notable region around the pre-trained checkpoint characterized by low disentanglement error. This finding echoes the ability of T5 to perform task arithmetic as demonstrated in Ilharco et al. [39] (Appendix D.6), thereby reinforcing the robustness of our conclusions. We will report this result in the revised version.

---

> > ### Comment · Reviewer_9cnX · 2023-08-15
> >
> > I thank the authors for their detailed feedback. After carefully reading through the clarification as well as the other reviews, most of my concerns have been addressed, and these new experimental results further strengthen the presented analyses. I thus raised my score.

---

### Official Review · Reviewer_sE5a · 2023-07-04

**Soundness:** 4 excellent
**Presentation:** 4 excellent
**Contribution:** 4 excellent
**Rating:** 9
**Confidence:** 4

**Summary:**

This paper theoretically and emperically investigates the reasons why task arithmetic (an emerging technique for editing pre-trained neural networks) works.
The paper shows that, contrary to previous hypotheses [39,79,80], lienarity of the fine-tuning on individual tasks is not sufficient to fully explain the success of task arithmetic (Sec 3).
Instead, the authors propose the related and straight-forward idea of weight disentaglement (Eq 4) as an explanation (Sec 4), also providing a measure of the disentaglment error for two tasks (Eq 6).
This idea is leveraged to improve task arithmetic performance by constraining the fine-tuning to linearized models - which importantly is shown to be an improvement over post-hoc linearlization, with marginal additional computational complexity (Sec 5).
Finally, an additional connection between task arithmetic and the eigen space of the Neural Tangent Kernel is used to argue that weight disentanglement (and hence the capacity for task artihmetic) are a learned properties during pre-training, not inherent properties of parameterization or architecture (Sec 6).
The empirical results focus on the CLIP image-text Vision Transformer model family, and many experimental details are reported, inclduing in the additional material in the appendix.

**Strengths:**

This paper is the best kind of NeurIPS paper; Beautifully written and a delight to read, the authors have considered a timely and pertinent problem in an emerging machine learning domain, applied a methodological theoretical investigation; clarified prior hypotheses in the literature; used the theoretical findings to propose a simple novel methodology, and compellingly evidenced the subsequent effectiveness with an appropriate evaluation protocol. As a cherry on the cake, the methodology is a 'drop in' method that can be applied easily to existing approaches, and actual code is provided without breaking review anonymity in the supplementary material. Bravo.

### Originality

 * There are multiple practical take-aways that are immediately useful; the task disentanglement error metric (Eq 6), intuition for why task vector coefficients should be << 1 (Ln 181), simple drop-in code in the appendix that enables re-production and immediate application of the ideas (Listing 1 in the appendix).

### Quality

 * The work appears to be of high quality, the metrics and emprical appear to support the theoretical findings and claims.

### Clarity

 * This paper is superbly well written, and the logical argument and flow is well constructed. This paper was a delight to read and think about.

### Signifigance

 * Multiple compelling and genuine avenues for future research are identfidied (Ln 239, Ln 286, Ln 315, Ln354)
 * Provides timely and much-needed (i) overview of the emerging and rapidly evolving technique of task arithmetic with pre-trained models, and (ii) the beginnings of a theoretical grounding and understanding for why this technique is possible.


**Weaknesses:**

* The main weakness of the paper in the present form is that the empirical results are limited to ViT models (specifically, CLIP). Within this modality, the empirical results are compelling (e.g. using multiple model families and 8x distinct datasets/tasks), and provide evidence for the theoretical findings, however the paper's authority and strength would be enhanced by demonstrating some of the key findings hold with another architecture or data type. E.g. I would love to see the results in Table 3 (Sec 6.2) replicated, even in a small way, with GPT or BERT on a text dataset. This would provider further strength for the empirical claims around Eigenfunction localization.


**Questions:**

 1. Ln 336 on 'Feature disentanglement'; to what degree is feature disentanglement enabled by over-parameterization / very large numbers of parameters? If this is the case, what is the parameter count relative to - i.e. some measure of the complexity of the input space or the function space represented by the data? Is there a way to quantify with a metric the 'capacity' for feature disentaglement of a given NN archiecture? These could be useful areas for investigation.
 2. Related to the above point, Ln 233 notes that the advantage of the linearized model task arithmetic approach diminishes as the number of model parameters increases. I find this curious and would have expecte the opposite result - do you have any intuition why this is the case?
 3. Pseudo-code or actual PyTorch code for Eq 6 would be a very helpful (and I assume straight-forward) addition to your supplementary material - this would be of immediate practical help to other researchers working on practical task arithmetic applications.
 4. Related to the above, Ln 297 - can the degree of local linear independence of the NTK eigenfunctions be measured/computed with a simple metric to test this condition in other models / datasets? This would likewise be a useful contribution of the paper.

**Limitations:**

 * It's not clear to me if there are situations where the linerarization technique won't be readily applicable. E.g. are some activation functions or more esoteric neural network architectures (e.g. recurrent NN's; long-term memory components) going to problematic to linearize? Some negative examples of where NTK linearization can't be used and/or approximations or work-arounds could be a useful addition in this regard.

---

> ### Author Rebuttal · Authors · 2023-08-10
>
> We really appreciate the reviewer’s enthusiasm and acknowledgment of the significance of our work and their engagement to improve it!  Below, we address their comments.
>
> **Generality of our results beyond CLIP/ViT models**
>
> We would like to emphasize that all our theoretical results are directly applicable to any model which satisfies Property 1 (Task arithmetic) regardless of pre-training scheme or architecture. Notably, any model satisfies Property 1 if and only if it is weight disentangled. As suggested, in order to show that our results are robust to the architecture choice and input modality, we have replicated our experimental analysis using convolutional neural networks and large language models:
>
> - We have repeated all our experiments using a CLIP ConvNeXt model pre-trained on LAION-400M. Remarkably, also for this architecture weight disentanglement is stronger in the tangent space to the pre-trained checkpoint (see left panels in Figure R.1 of the *Author Response document* attached [here](https://openreview.net/forum?id=0A9f2jZDGW&noteId=7E6o5YEkJw)) and linearized fine-tuning enhances task arithmetic (see Table R.1 of the *Author Response document*). We will present the complete set of new results in the revised version.
>
> - In addition, in order to investigate the effects of pre-training, we are repeating the same analysis for non-contrastively pre-trained (closed-vocabulary) ViT and ConvNeXt models, and we will add these results to the final version as well. As suggested, we will also make explicit what architectures we consider, both in the abstract and in the main text.
>
> - To substantiate the generality of weight disentanglement, we conducted a new experiment on a pre-trained T5-Base model from Hugging Face, fine-tuned on two benchmark NLP tasks (sentiment analysis on movie reviews and question answering). The results, illustrated in the right panel in Figure R.1 of the *Author Response document*, show a notable region around the pre-trained checkpoint characterized by low disentanglement error. This finding echoes the ability of T5 to perform task arithmetic as demonstrated in Ilharco et al. [39] (Appendix D.6), thereby reinforcing the robustness of our conclusions. We will report this strong result in the paper. Finally, while exploring whether linearization improves weight disentanglement and task arithmetic for different modalities – such as language – is undoubtedly captivating, it goes beyond the current scope of the paper, and we reserve it for subsequent investigations.
>
> **Disentanglement and overparameterization**
>
> We are not aware of any results showing to what extent feature disentanglement is enabled by overparameterization. Intuitively, achieving a distinct disentanglement in feature space, as well as in weight space in the case of weight disentanglement, demands a sufficient number of model parameters. A systematic study of these quantities as the model size scales lies beyond our current scope. However, we concur that this is a fascinating question for future research.
>
> **Linear vs non-linear**
>
> Our intuition for the observation of the diminishing advantage of linearized-model task arithmetic with increasing model size is that larger models tend to stay closer to the NTK approximation. In other words, they tend to behave linearly without being explicitly constrained to do so (see, e.g., Figure 2). One explanation for this is that larger models, being more over-parameterized, inherently induce a stronger kernel behavior during fine-tuning, i.e., having more parameters, each parameter requires smaller adjustments to fit the training examples. As a result, being closer to their linearized counterparts, larger models have better weight disentanglement and can perform task arithmetic similarly to linearized models (see also the answer Effect of Scale on Non-Linear Advantage to reviewer [DjkZ](https://openreview.net/forum?id=0A9f2jZDGW&noteId=PEwcdQxgup) and the new results in Section 3 of the *Author Response document* displaying weight disentanglement as a function of model size).
>
> **Pseudo-code for the weight disentanglement error**
>
> We thank the reviewer for their suggestion of providing the PyTorch code for computing the disentanglement error in the Appendix. This addition will enhance reproducibility, and we intend to implement it in the revised version.
>
> **Eigenfunction localization**
>
> Approximating the eigenfunctions of the NTK is a costly operation since it requires computing the kernel matrix and diagonalizing it. Hence, in general, measuring localization or other properties of the eigenfunctions is challenging. Moreover, in practice, all these properties are not displayed exactly, so a sound investigation should take into account this fact as well. All in all, this precludes an exhaustive exploration within the confines of this paper. Yet, we agree and believe that studying the spectral properties of the NTK for different datasets, architectures, and modalities holds promise for future research. We will acknowledge the importance of this avenue in our manuscript, reflecting your input.
>
> **Linearization of other architectures**
>
> Linearization readily works with the majority of architectures currently used, encompassing convolutional and transformer architectures with both smooth and non-smooth activation functions (as shown by our new experiments using ConvNeXts). Although our current framework doesn't explicitly address recurrent architectures, we believe that implementing linearization for recurrent architectures is possible We will explicitly outline the range of applicability of our procedure in Appendix B (Implementation aspects of linearized models).
>
> We thank the reviewer for their valuable feedback and remain available to answer further questions or provide more clarifications regarding the previous points.

---

> > ### Comment · Reviewer_sE5a · 2023-08-15
> > **Acknowledgement of author responses**
> >
> > I thank the authors for their responses and the further experiments. These results further strengthen what is already a great paper. Well done.

---

### Official Review · Reviewer_DjkZ · 2023-07-06

**Soundness:** 4 excellent
**Presentation:** 4 excellent
**Contribution:** 4 excellent
**Rating:** 8
**Confidence:** 4

**Summary:**

This paper presents a comprehensive analysis of task arithmetic using pre-trained CLIP models. It challenges the early hypothesis that task arithmetic arises from linear fine-tuning in the NTK regime and introduces weight disentanglement as a necessary condition for enabling task arithmetic.  Further experiments demonstrate that linearized fine-tuning of pre-trained CLIP models exhibits stronger weight disentanglement and improved task arithmetic compared to standard, non-linear fine-tuning. This result is accompanied by an analysis of the NTK spectrum to facilitate the understanding of weight disentanglement in linearized models. Lastly, empirical evidence suggests that weight disentanglement emerges from large-scale pre-training. The findings of the paper may shed new light on the effective adaptation of foundation models for downstream applications.

**Strengths:**

- This paper advances the theoretical understanding of task arithmetic. In particular, it introduces weight disentanglement as a strong indicator of task arithmetic, and demonstrates that linearized fine-tuning in the tangent space of pre-trained weights promotes weight disentanglement. It further analyzes the NTK spectrum of linearized models and presents a sufficient condition for weight disentanglement.

- In the meantime, the findings of the paper are significant on the practical side. The experiment results justify an emerging fine-tuning scheme for adapting foundation models.

- Overall, this paper strikes a good balance between theory (NTK) and practice (fine-tuning pre-trained models). Embracing a broad audience is a key strength of the paper.

- Finally, the paper is very well-written. The flow of presentation is very easy to follow.

**Weaknesses:**

- Overall, I found the theoretical analysis motivating and the experiment results convincing. That said, all conclusions of the paper are drawn from CLIP fine-tuning, which makes me wonder whether the same findings are valid for other pre-trained models that similarly exhibit task arithmetic. To this end, I encourage the authors to report results on a second pre-trained model that differs in network architecture (e.g., ResNet), learning objective (e.g., MAE) or input modality (e.g., natural language), in order to establish the generality of their findings.

- The authors observed that increasing model size (ViT-B/32->ViT-L/14) closes the gap between linearized and non-linear fine-tuning. Is it a consequence of stronger weight disentanglement because the fine-tuning of larger models approaches the NTK regime? Visualization of weight disentanglement throughout the paper only considers the smallest ViT-B/32 model. Similar visualizations for larger models could be informative.

**Questions:**

These questions are likely outside the scope of the paper, yet I think addressing any of them via an empirical analysis could strengthen the work.

- Task arithmetic makes the strong assumption that datasets from different tasks have disjoint support. This is often not true in practice. Does task addition still yield cooperative behavior when datasets overlap? How does linearized fine-tuning compare to standard, non-linear fine-tuning, especially when the assumption of non-overlapping data is violated?

- In linearized fine-tuning, the optimization is restricted to the tangent space of pre-trained weights. Notably, this is conceptually similar to LoRA and adaptor-based fine-tuning, where a task vector exhibits low-dimensional structure. A natural question to ask is whether these fine-tuning approaches also produce favorable weight disentanglement / task arithmetic.

**Limitations:**

The paper discussed two main limitations. First, linearized fine-tuning is currently implemented using the JVP algorithm, which doubles the cost of a forward pass as compared to standard fine-tuning. Second, the spatial localization of NTK eigenfunctions is a sufficient (yet not necessary) condition for enabling task arithmetic. While the linearized models indeed respond to disjoint spatial regions in the experiments, this is not a must for task arithmetic to hold.

---

> ### Author Rebuttal · Authors · 2023-08-10
>
> We appreciate that the reviewer recognized that our paper is well-written, our analysis is motivating, and our experiments are convincing and their engagement to improve it. Below, we address their comments.
>
> **Generality of our results beyond CLIP/ViT models**
>
> We would like to emphasize that all our theoretical results are directly applicable to any model which satisfies Property 1 (Task arithmetic) regardless of pre-training scheme or architecture. Notably, any model satisfies Property 1 if and only if it is weight disentangled. However, in order to show that our results are robust to the architecture choice and input modality, we have replicated our experimental analysis using convolutional neural networks and large language models:
>
> - We have repeated all our experiments using a CLIP ConvNeXt model pre-trained on LAION-400M. Remarkably, also for this architecture weight disentanglement is stronger in the tangent space to the pre-trained checkpoint (see left panels in Figure R.1 of the *Author Response document* attached [here](https://openreview.net/forum?id=0A9f2jZDGW&noteId=7E6o5YEkJw)) and linearized fine-tuning enhances task arithmetic (see Table R.1 of the *Author Response document*). We will present the complete set of new results in the revised version.
>
> - In addition, in order to investigate the effects of pre-training, we are repeating the same analysis for non-contrastively pre-trained (closed-vocabulary) ViT and ConvNeXt models, and we will add these results to the final version as well. As suggested, we will also make explicit what architectures we consider, both in the abstract and in the main text.
>
> - To substantiate the generality of weight disentanglement, we conducted a new experiment on a pre-trained T5-Base model from Hugging Face, fine-tuned on two benchmark NLP tasks (sentiment analysis on movie reviews and question answering). The results, illustrated in the right panel in Figure R.1 of the *Author Response document*, show a notable region around the pre-trained checkpoint characterized by low disentanglement error. This finding echoes the ability of T5 to perform task arithmetic as demonstrated in Ilharco et al. [39] (Appendix D.6), thereby reinforcing the robustness of our conclusions. We will report this strong result in the paper. Finally, while exploring whether linearization improves weight disentanglement and task arithmetic for different modalities – such as language – is undoubtedly captivating, it goes beyond the current scope of the paper, and we reserve it for subsequent investigations.
>
> **Effect of scale on non-linear advantage**
>
> The reviewer correctly highlights that by scaling the number of model parameters, the performance of linearized fine-tuning becomes closer to the one of standard non-linear fine-tuning. To the best of our knowledge, this is a novel observation.  As commented briefly in Appendix D.1, one plausible interpretation is that larger models, which are more over-parameterized, inherently induce a stronger kernel behavior during fine-tuning. Namely, since the models have more parameters, each parameter has to change less to fit the training examples. As a result, they tend to stay closer to the NTK approximation, closing the gap with linearized models and taking benefit of the better weight disentanglement of the models lying in the tangent space.
>
> In response to the reviewer’s suggestion, we conducted supplementary experiments that visualize how weight disentanglement varies changing the model scale (see Figure R.2 of the *Author Response document*). Consistently to our results, larger models exhibit stronger weight disentanglement, as highlighted by the larger light region in the right panel of the first row of Figure R.2. Yet, interestingly, the models linearly fine-tuned are always more weight disentangled than the non-linearly fine-tuned ones, highlighting the strength of linearized models for model editing. We will add this discussion and the new results to the paper.
>
> **Open questions**
>
> We thank the reviewer for raising these stimulating questions, although we agree that these are beyond our current scope. In particular, the disjoint support assumption adopted in our work stems from the way in which task arithmetic was first introduced in Ilharco et al. [39], wherein the datasets are indeed disjoint. Yet, we acknowledge that studying task arithmetic in other settings is an interesting direction (see also the answer Disjoint Task Support Hypothesis to Reviewer [eVrq](https://openreview.net/forum?id=0A9f2jZDGW&noteId=QWXTFZW72i)). Similarly, studying the sparsity of task vectors and the effects of LoRA training are exciting open questions and avenues that warrant separate exploration.
>
> We thank the reviewer for their valuable feedback and remain available to answer further questions or provide more clarifications regarding the previous points.

---

> ### Comment · Reviewer_DjkZ · 2023-08-15
> **Updated rating**
>
> The rebuttal addressed my concerns. The new results on a convolutional backbone and NLP tasks empirically confirmed the generality of the analysis. Overall, this work is well-positioned to inspire both the theory and practice of foundation model adaptation. I thus raised my rating from weak accept to strong accept.

---

### Official Review · Reviewer_qb7d · 2023-07-07

**Soundness:** 4 excellent
**Presentation:** 4 excellent
**Contribution:** 3 good
**Rating:** 10
**Confidence:** 4

**Summary:**

This work challenges the prevailing belief regarding the origin of task arithmetic in CLIP models. While it is commonly attributed to the linear nature of fine-tuning, the authors argue that the critical factor lies in the "weight disentanglement" that occurs between tasks during the fine-tuning process. The paper presents a theoretical analysis and extensive empirical validation to support this claim.

**Strengths:**

This paper explores an intriguing topic with promising real-world applications. The authors exhibit strength in establishing a solid theoretical foundation, formulating straightforward research questions, and providing well-justified explanations that strike a remarkable balance between accessibility and rigor.

The content is commendably clear and understandable, facilitating understanding for readers from diverse backgrounds. Personally, I was able to (at least, I think) grasp portions of the paper (Section 6) which required prerequisite knowledge I was not already familiar with.

Notably, this paper achieves a synergy between its experimental design and the overarching claim that "Task arithmetic is not merely a result of linear fine-tuning," but rather depends on "weight disentanglement of the model with respect to the fine-tuning task set." The authors successfully validate the performance of their proposed method through rigorous experiments (and ablations) while simultaneously providing substantial support for key theoretical hypotheses.

Overall, this paper represents a remarkable contribution, embodying the meticulousness, clarity, and scholarly standards I would expect in top-tier NeurIPS submissions.

I’m starting my recommendation with an 8, given my not-high confidence, I’ll adjust it according to the rebuttal discussion.

**Weaknesses:**

One aspect that requires attention is the clarification of the paper's applicability beyond CLIP models. Although there are specific references throughout the text (e.g., lines 38-49 in the introduction) emphasizing the focus on CLIP models, the overall messaging may appear too general regarding the broader applicability of the proposed method and study. To address this, I recommend at least modifying the abstract to explicitly state the paper's objective as "… a comprehensive study of task arithmetic in **CLIP** models…" instead of using the term "vision-language models" which could be misleading. Furthermore, exploring preliminary tests or investigations concerning the weight disentanglement of different pre-trained models would be valuable. For instance, considering the citation of various architectures in Section 6.2 (lines 301-303), extending the analysis to include some of them (e.g., convolutional neural networks) would be advantageous, strengthening the paper's applicability and relevance to a broader range of models and areas.

Additionally, while Tables 1 and 2 provide average performance across tasks, it would be beneficial to include the standard deviation to provide a more comprehensive understanding of the variability in performance and, therefore, the method's robustness.

**Questions:**

I’ve seen that in the supplementary material, there are some code snippets, but do the authors plan to release the full codebase upon acceptance?

**Limitations:**

The authors adequately addressed the limitations.

---

> ### Author Rebuttal · Authors · 2023-08-10
>
> We sincerely appreciate the reviewer’s recognition of our work and their engagement to improve it!  In what follows, we address their comments.
>
> **Generality of our results beyond CLIP/ViT models**
>
> We would like to emphasize that all our theoretical results are directly applicable to any model which satisfies Property 1 (Task arithmetic) regardless of pre-training scheme or architecture. Notably, any model satisfies Property 1 if and only if it is weight disentangled. However, in order to show that our results are robust to the architecture choice and input modality, we have replicated our experimental analysis using convolutional neural networks and large language models:
>
> - We have repeated all our experiments using a CLIP ConvNeXt model pre-trained on LAION-400M. Remarkably, also for this architecture weight disentanglement is stronger in the tangent space to the pre-trained checkpoint (see left panels in Figure R.1 of the *Author Response document* attached [here](https://openreview.net/forum?id=0A9f2jZDGW&noteId=7E6o5YEkJw)) and linearized fine-tuning enhances task arithmetic (see Table R.1 of the *Author Response document*). We will present the complete set of new results in the revised version.
>
> - In addition, in order to investigate the effects of pre-training, we are repeating the same analysis for non-contrastively pre-trained (closed-vocabulary) ViT and ConvNeXt models, and we will add these results to the final version as well. As suggested, we will also make explicit what architectures we consider, both in the abstract and in the main text.
>
> - To substantiate the generality of weight disentanglement, we conducted a new experiment on a pre-trained T5-Base model from Hugging Face, fine-tuned on two benchmark NLP tasks (sentiment analysis on movie reviews and question answering). The results, illustrated in the right panel in Figure R.1 of the *Author Response document*, show a notable region around the pre-trained checkpoint characterized by low disentanglement error. This finding echoes the ability of T5 to perform task arithmetic as demonstrated in Ilharco et al. [39] (Appendix D.6), thereby reinforcing the robustness of our conclusions. We will report this strong result in the paper. Finally, while exploring whether linearization improves weight disentanglement and task arithmetic for different modalities – such as language – is undoubtedly captivating, it goes beyond the current scope of the paper, and we reserve it for subsequent investigations.
>
> **Robustness of our method**
>
> In regard to the variability in performance in Tables 1 and 2, it is important to consider that the diverse nature of the tasks results in distinct levels of difficulty. Hence, standard deviations are likely more affected by this variability in task difficulty than the robustness of a given method. Nevertheless, we concur with the reviewer that solely looking at averages might not convey the whole picture. To address this, we showed that the improvements in performance are consistent across tasks in Appendix D.2. This clarification will be further stressed in the revised version.
>
> **Code release**
>
> We confirm that we will release the complete codebase in a public GitHub repository once the work undergoes deanonymization.
>
> We thank the reviewer for their valuable feedback and remain available to answer further questions or provide more clarifications regarding the previous points.

---

> > ### Comment · Reviewer_qb7d · 2023-08-19
> >
> > I want to thank the authors for their thorough response to every reviewer and for addressing all of my concerns.
> >
> > Having closely examined the other reviews and the authors' responses, it's clear they've put in substantial effort to incorporate reviewer feedback and improve their work (which was already of high level). During the rebuttal period, the authors added more experiments and clarifications that strengthened the paper's contributions and impact across fields, significantly extending its applicability to different models and fields.
> >
> > Given the unanimous acclaim for the paper's strengths (i.e., exceptional clarity in writing, substantial theoretical and experimental components, and valuable implications) and the authors' diligent revisions, I am confident in upgrading my recommendation to a score of 10.

---

### Author Rebuttal · Authors · 2023-08-10

We kindly thank all the reviewers for their time and for providing valuable feedback on our work. We appreciate that reviewers have pointed out that our work is interesting (Reviewer [eVrq](https://openreview.net/forum?id=0A9f2jZDGW&noteId=kbSkLPUU32)), intriguing (Reviewer [qb7d](https://openreview.net/forum?id=0A9f2jZDGW&noteId=MpKbXwmZKw)), and very well written (Reviewers [qb7d](https://openreview.net/forum?id=0A9f2jZDGW&noteId=MpKbXwmZKw), [DjkZ](https://openreview.net/forum?id=0A9f2jZDGW&noteId=v6Lz3dFhri), [sE5a](https://openreview.net/forum?id=0A9f2jZDGW&noteId=CDcYUaXFFc)), and that our results are solid (Reviewer [qb7d](https://openreview.net/forum?id=0A9f2jZDGW&noteId=MpKbXwmZKw)), impressive (Reviewer [9cnX](https://openreview.net/forum?id=0A9f2jZDGW&noteId=CetQ5EK9GW)), and impactful (Reviewers [DjkZ](https://openreview.net/forum?id=0A9f2jZDGW&noteId=v6Lz3dFhri), [qb7d](https://openreview.net/forum?id=0A9f2jZDGW&noteId=MpKbXwmZKw)).

In response to the reviews, we ran a series of **new experiments** to show the generality of our findings. Specifically,
- We have replicated our experimental analysis using a **convolutional architecture**. Our new results, reported in Table R.1 and Figure R.1 of the *Author Response document* (see attached pdf) reveal that also for this architecture weight disentanglement is stronger in the tangent space and linearized fine-tuning enhances task arithmetic.
- We have extended the experimental results on weight disentanglement to language by
demonstrating that a T5-Base model, fine-tuned on two distinct **NLP tasks**, exhibits a region around its pre-trained initialization with low weight disentanglement error (Figure R.1, right panel).
- We have analyzed the **effect of model scale** on weight disentanglement, showing that larger models are more weight disentangled, but not as much as their linearized counterparts (Figure R.2)

We hope that these new results and the clarifications detailed in the individual comments given to each reviewer will effectively address the concerns raised during the review process. We remain available for engaging in any further discussions that may arise, and we thank you once again for your comments.

---

### Decision · Program_Chairs · 2023-09-21

**Decision:**

Accept (oral)

**Comment:**

This paper studies the use of task vectors, i.e. the difference between a fine-tuned model and a pre-trained model, to perform task arithmetic, i.e. adding or removing capabilities to a model. A new theoretical viewpoint is developed that links the success of task vectors to "weight disentanglement", and this perspective is used to develop a new method for performing task arithmetic that outperforms prior methods. Reviewers all appreciated the clearly articulated theory and convincing experiments and unanimously voted to accept the paper.